# Mixing Expertise with Confidence:
# A Mixture of Experts Framework for Robust Multi-Modal Continual Learning

Md Abdullah Al Forhad [1]  Yuansheng Zhu [2]  Abhinab Acharya [2]  Xumin Liu [2]  Qi Yu [2]  Weishi Shi [1]

## Abstract

The Mixture of Experts (MoE) framework is widely used in continual learning to mitigate catastrophic forgetting. MoEs typically combine a small inter-task shared parameter space with largely independent expert parameters. However, as the number of tasks increases, the shared space becomes a bottleneck, reintroducing forgetting, while fully independent experts require explicit task ID predictors (e.g., routers), adding complexity. In this work, we eliminate the inter-task shared parameter space and the need for a task ID predictor by enabling expert communication and allowing knowledge to be shared dynamically, akin to human collaboration. We bridge the inter-expert knowledge sharing by leveraging the open-set learning capabilities of a multimodal foundation model (e.g., CLIP), thereby providing "expert priors" that bolster each expert's task-specific representations. Guided by these priors, experts learn calibrated inter-task posteriors. Additionally, multivariate Gaussians over the learned posteriors promote complementary specialization among experts. We propose new evaluation benchmarks that simulate realistic continual learning scenarios, and our prior-conditioned strategy consistently outperforms existing methods across diverse settings without relying on reference datasets or replay memory.

## 1. Introduction

Continual learning (CL) emerges as a highly realistic and significant problem in pursuing Artificial General Intelligence (De Lange et al., 2022). Unlike classical supervised learning, which operates within fixed problem boundaries and assumes access to static, independent, and identically distributed (i.i.d.) datasets, continual learning (CL) emphasizes the need for models to adapt to new tasks while being restricted from accessing past-task data. The core objective of continual learning (CL) is to enable models to generalize across previously encountered tasks and gain the ability to solve a broad range of learning challenges over time.

While early research in CL mainly focused on single-modal data, modern machine learning increasingly relies on multimodal resources, such as vision-language or vision-audio data (Zheng et al., 2023; Yu et al., 2024). This shift demonstrates multimodal data's growing prevalence and importance in real-world applications. Developing continual learning methods to handle such data is critical to ensure robust performance across diverse downstream tasks. In addition, multimodal learning offers promising solutions to some inherent challenges single-modal continual learning faces. For example, a typical failure case in single-modal learning happens when an image contains overlapping concepts among different tasks. Consider an image of a flower on a table; its label might be "flower" in one task and "table" in a later task. A single-modal learner relying solely on image features will overfit to the most recently learned task, connecting the learned features of the image with "table" and forgetting about the label "flower". On the other hand, multimodal data provides textual descriptions like "Flowers on the table" and "A table with flowers on its top" to help the model disentangle specific features. Aligning visual and textual information helps models learn robust, meaningful features and retain task-specific knowledge despite domain shifts. We analyze this behavior in Appendix E.

Furthermore, the complementary nature of multimodal data enables robust zero-shot learning, as exemplified by CLIP (Radford et al., 2021), which shows strong generalization from large-scale image-text pairs (Thengane et al., 2022). This capability is particularly valuable in CL scenarios where initiating a new learning phase often involves limited training data. Multimodal learners can leverage these zero-shot abilities to kick-start new tasks with a reduced risk of catastrophic forgetting (CF), a common phenomenon

---

[1]Department of Computer Science and Engineering, University of North Texas, Denton, TX, USA [2]Rochester Institute of Technology, Rochester, NY, USA. Correspondence to: Md Abdullah Al Forhad <MdAbdullahAl.Forhad@unt.edu>, Weishi Shi <Weishi.Shi@unt.edu>.

*Proceedings of the 43rd International Conference on Machine Learning*, Seoul, South Korea. PMLR 306, 2026. Copyright 2026 by the author(s).

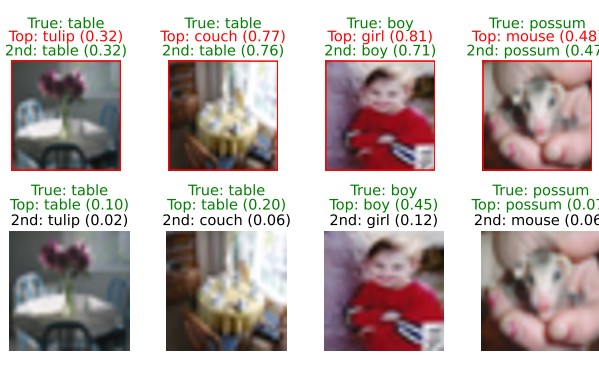

*Figure 1.* Comparison of independently trained experts (top) vs. our MoE design (bottom) on CIFAR-100 (10 steps). **Top**: OOD experts exhibit overconfidence despite incorrect predictions, leading to erroneous task identification. For instance, in the second image from right, an expert trained in "boy" images shows less confidence than one who has never seen such images. **Bottom**: our MoE yields the correct task ID and prediction. See Appendix C.3 for details.

of continual learning (French, 1999; McCloskey & Cohen, 1989). Even with only single-modal data, incorporating multimodal frameworks can enhance learning outcomes. By leveraging large pre-trained generators to create complementary modalities (e.g., pairing visual data with textual descriptions), models gain richer inputs and benefit from the performance boost of multimodal learning. Although multimodal models exhibit strong zero-shot learning capabilities, relying solely on this strength is insufficient to address the core challenges of CL. CL not only requires models to achieve reasonable performance on new tasks but also demands a delicate balance between improving task-specific performance and preserving previously acquired knowledge. Several recent methods, such as parameter regularization and parameter-efficient fine-tuning, have attempted to address these challenges by enhancing the adaptability of multimodal models like CLIP across tasks (Zheng et al., 2023; Yu et al., 2024). These approaches aim to retain prior knowledge while fine-tuning for new tasks. However, in practice, they often overfit the current task learning objective, overwriting knowledge from previous tasks, resulting in catastrophic forgetting (CF). This outcome should not be surprising. While multimodality can sometimes mitigate CF, such as in our earlier example, where textual descriptions provided sufficient complementary information to guide task-specific feature learning, this is not universally effective. Overfitting and CF remain significant challenges when the modalities involved fail to provide adequate signals to distinguish between task-specific patterns or when the model's tunable parameters are insufficient to support effective disentanglement. In these cases, the complementary nature of multimodal data alone is insufficient to resolve the broader issues inherent in CL, particularly in balancing task-specific learning with knowledge retention.

To tackle these limitations, we adopt the Mixture of Experts

(MoE) architecture (Yu et al., 2024; Ma et al., 2024; Wu et al., 2024; Rypeść et al., 2023) as a robust framework for multimodal CL. A key advantage of MoE is its divide-and-conquer design, which allocates separate learning spaces for different tasks, thereby mitigating CF by preventing direct interference between task-specific knowledge. However, conventional MoE implementations rely on shared parameters as a communication mechanism between experts. While this design benefits static learning setups, it becomes problematic in a CL setting. The low-level and hard-to-control information flow between experts allows "shortcut features" from previous tasks to propagate unchecked, distorting the learning of new tasks. For example, suppose the first expert is trained to classify "cow". During learning, it not only captures features specific to cows but also stores background cues like grass as a useful feature within the shared parameters. When a second expert is later introduced to learn "horse", it inherits these shared parameters and shortcuts the learning process by associating grass with the "horse" label rather than focusing on the distinct features of horses. This uncontrolled transfer of shortcut features distorts task-specific learning and increases the risk of overconfident false positives in CL.

To address the abovementioned issues, we propose modifying the MoE design by entirely removing shared parameters and enabling expert communication through a dynamic temperature scaling mechanism based on expert predictions. This approach ensures that new experts actively incorporate the opinions of past experts when learning from current task data rather than passively inheriting signals from a shared backbone. As a result, our design prevents harmful biases from being implicitly transferred while allowing task-specific learning to remain adaptive. As shown in Figure 1, this significantly reduces overfitting to shortcut features and mitigates the issue of overconfident classifications. Additionally, leveraging the strengths of multimodal models in out-of-distribution (OOD) detection, we introduce distribution-aware multivariate Gaussian-based weighting, which enhances expert coordination during inference. Our experiments demonstrate that our method consistently outperforms existing approaches across various benchmarks. In summary, this paper advances CL in multimodal models through the following contributions:

- We propose a novel expert collaboration scheme based on dynamic confidence-aware temperature-scaling, which enables experts to interact without parameter sharing. This approach provides a memory-free continual learning framework with robust in-distribution (IND) and out-of-distribution (OOD) confidence calibration, without requiring reference datasets or routers as prior methods do.

- We provide a formal analysis showing how our

confidence-aware temperature scaling introduces an implicit epistemic prior that improves predictive certainty in regions dominated by shortcut features.

- We enhance expert communication during inference via a distribution-aware weighting mechanism that regularizes and improves expert selection, boosting performance.

## 2. Related Work

**Continual Learning (CL)** in deep neural networks aims to mirror human lifelong learning while mitigating catastrophic forgetting (CF) (Chen & Liu, 2016). The main CL paradigms are **Class-Incremental Learning (CIL)** (Oren & Wolf, 2021; Belouadah & Popescu, 2019; De Min et al., 2023; Liu et al., 2023; Yan et al., 2021; Douillard et al., 2022), where the task-ID (TID) is unknown during inference (making it the most challenging (Chen & Liu, 2016)), and **Task-Incremental Learning (TIL)** (Mallya & Lazebnik, 2018; Oren & Wolf, 2021), where the TID is provided. To mitigate CF, researchers have explored regularization-based methods (Kirkpatrick et al., 2017; Zhu et al., 2021; Zeng et al., 2019), which constrain parameters across tasks; replay-based methods (Rebuffi et al., 2017; Ebrahimi et al., 2021; Buzzega et al., 2020; Boschini et al., 2022; Lopez-Paz & Ranzato, 2017), which reuse stored data from previous tasks; and parameter-isolation methods (Mallya et al., 2018; Serra et al., 2018; Kim et al., 2022; Wang et al., 2022a;b; Rusu et al., 2016; Aljundi et al., 2017; Rosenfeld & Tsotsos, 2020), which allocate distinct parameters for each task. More recently, the CL community has turned to **multimodality**, such as **Vision-Language Models (VLMs)**, which learn from both visual and textual inputs, enabling zero-shot transfer. In particular, CLIP (Radford et al., 2021) trains separate encoders for images and text using a contrastive objective and often outperforms traditional CL approaches with no additional training (Thengane et al., 2022). ZSCL (Zheng et al., 2023), which uses CLIP, employs a parameter-regularization fine-tuning strategy to mitigate CF, but CLIP's large parameter count poses computational challenges. To address this, **MoE-Adapters** (Yu et al., 2024) propose a **parameter-efficient fine-tuning** (Zhang et al., 2020; Karimi Mahabadi et al., 2021; Jia et al., 2022) framework using **Mixture-of-Experts (MoEs)** (Shazeer et al., 2017), although shared-parameter structure remains susceptible to CF. Moreover, ZSCL and MoE-Adapters rely on a reference dataset, limiting real-world applicability, while some task-specific methods still require TID during inference (Wen et al., 2020; Hung et al., 2019; Golkar et al., 2019; Fernando et al., 2017; Collier et al., 2020). Li et al. (2025) provides complementary theoretical analyses of MoEs in CL. Beyond these expert-based approaches, HiDe-PET (Wang et al., 2025) introduces a hierarchical decomposition for parameter-efficient tuning (PET). It decomposes the CL objective into within-task prediction (WTP), task-identity inference (TII), and task-adaptive prediction (TAP), and leverages PET techniques to explicitly optimize all three. However, they cannot achieve knowledge transfer due to task-specific parameters isolation, whereas our work introduces dynamic inter-expert communication. A more recent approach, Regression-based Analytic Incremental Learning (RAIL) (Xu et al., 2024) uses ridge regression on VLMs image embeddings. RAIL is limited in its use of only the pre-trained vision embeddings and requires memory. Another branch, **Cross-domain Task-Agnostic Incremental Learning (X-TAIL)**, addresses domain shifts across tasks (Xu et al., 2024). X-TAIL extends **Multi-Domain Task Incremental Learning (MTIL)** (Yu et al., 2024) by removing domain hints, which enables TIL multi-domain scenarios and evaluate zero-shot transfer. However, it does not comprehensively assess generalization performance on unseen subpopulations (Santurkar et al., 2021; Liang et al., 2022).

Our approach focuses on zero-shot transfer and robustness in multimodal CL, targeting realistic benchmarks such as sub-population shifts and traditional CL settings, without relying on `memory` or a `reference dataset`.

## 3. Methodology

**Preliminary.** We define the CL setting as follows. Consider a pre-trained multimodal foundation model $\mathcal{F} : \mathcal{X} \to \mathcal{Y}$ that learns across $T$ tasks and maps input $\mathcal{X}$ to output $\mathcal{Y}$. The tasks $\{\mathcal{T}^t\}_{t=1}^{T}$ arrive sequentially, with only one accessible at each timestamp $t$. Each task $\mathcal{T}^t = \{\mathcal{D}^t, \mathcal{C}^t\}$ includes a dataset $\mathcal{D}^t = \{(\mathbf{x}_i^t, y_i^t)\}_{i=1}^{|\mathcal{D}^t|}$, sampled *i.i.d.* from $\mathcal{P}_{\mathcal{X}_t \times \mathcal{Y}_t}$, and a set of classes $\mathcal{C}^t = \{c_j^t\}_{j=1}^{M^t}$, where $M^t$ is the number of classes in $\mathcal{D}^t$. In *class-incremental learning* (CIL), each task has disjoint label spaces ($\mathcal{Y}_i \cap \mathcal{Y}_j = \emptyset$, $\forall i \neq j$). In *domain-incremental learning* (DIL), tasks share the same labels ($\mathcal{Y}_i = \mathcal{Y}_j$) but have different input domains ($\mathcal{X}_i \neq \mathcal{X}_j$). Our approach generalizes across both settings.

To demonstrate a multimodal model as a continual learner, we will henceforth consider Contrastive Language-Image Pre-training (CLIP) (Radford et al., 2021) as a running example. CLIP, denoted as $\mathcal{F} = \mathcal{G}\{\mathbf{E}_{\text{visual}}, \mathbf{E}_{\text{text}}\}$, uses two parallel encoders: a Transformer-based text encoder $\mathbf{E}_{\text{text}}$ and an image encoder $\mathbf{E}_{\text{visual}}$ (e.g., Vision Transformer). Both produce embeddings of the same dimension, mapping images and text into a joint embedding space.

To fine-tune the model on a given task $\mathcal{T}^t$, each class category $c_j^t$ is transformed via a prompt template (e.g., `"a photo of a {category}"`), and the text encoder $\mathbf{E}_{\text{text}}$ yields text embeddings $\{\mathbf{t}_j^t\}_{j=1}^{M^t}$. For an image $\mathbf{x}_i^t$, the visual encoder $\mathbf{E}_{\text{visual}}$ produces $\mathbf{v}_i^t$. Cosine similarities $s_{i,j}^t = \text{sim}(\mathbf{t}_j^t, \mathbf{v}_i^t)$ are calculated on the joint embedding space. Finally, cross-entropy loss is calculated in the form

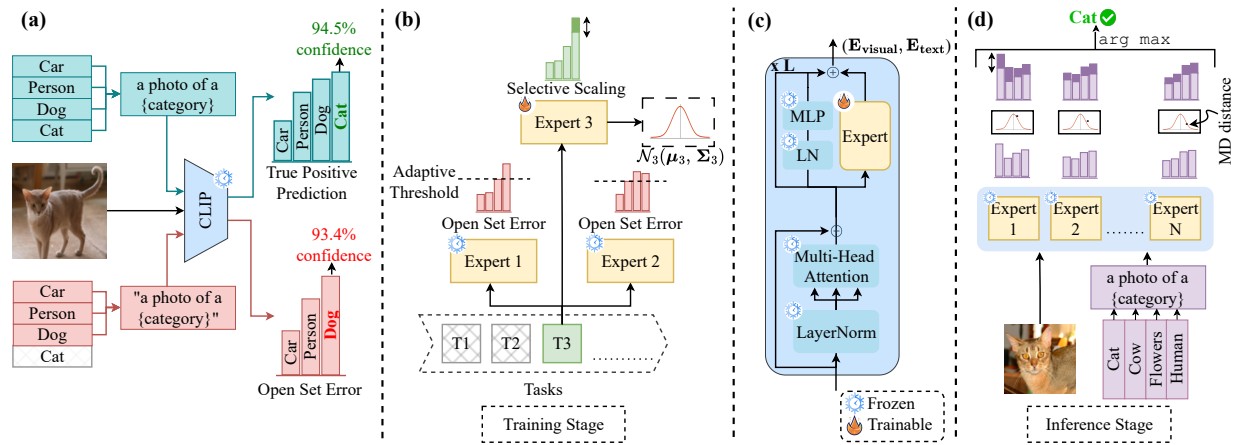

*Figure 2.* **(a)** CLIP confidently predicts the correct class "cat" when the true label is present, but makes a high-confidence mistake (open-set error) when the "cat" label is absent. **(b)** training: open-set errors from previous experts filter semantically similar samples (Equation 3), and selective scaling (shown as **darker green**) calibrates the current expert (Equation 4). The expert also learns a multivariate Gaussian from the vision embeddings (Equation 7). **(c)** expert architecture as adapters that support residual updates by caching frozen activations, enabling adapter updates without full backbone recomputation in order to enable efficient inference. **(d)** inference: MD offers sample-specific guidance (**darker purple** denotes weight) (Equation 8).

of equation 1 where $L$ is CLIP's logit scaling parameter.

$$\mathcal{L}^t = \frac{1}{|\mathcal{D}^t|} \sum_{i=1}^{|\mathcal{D}^t|} \mathcal{L}_{\text{CE}}\left(\boldsymbol{\theta}^t; \mathbf{x}_i^t, y_i^t\right) ; \text{ where } \mathcal{L}_{\text{CE}}\left(\boldsymbol{\theta}^t; \mathbf{x}_i^t, y_i^t\right)$$

$$= -\sum_{j=1}^{M^t} \left[ y_{i,j}^t \ln \left( \frac{\exp\left(L \cdot s_{i,j}^t\right)}{\sum_{k=1}^{M^t} \exp\left(L \cdot s_{i,k}^t\right)} \right) \right] \tag{1}$$

### 3.1. Mixture-of-Experts Design.

The Mixture-of-Experts (MoE) (Jacobs et al., 1991) framework has been widely adopted by the research community. In MoE, each expert is a distinct neural network with independent inputs and outputs, potentially sharing or maintaining separate parameters (Shazeer et al., 2017; Chen et al., 2023). The primary goal is to enable each expert to learn unique perspectives or handle different tasks. As discussed in Section 1, we use task-specific experts to facilitate two benefits: (1) maximizing task-specific performance and (2) facilitating incremental expansion as new tasks arise.

Although MoE offers modularity, assigning full models to each expert leads to growing parameter overhead as tasks accumulate. We follow prior strategies like LoRA (Hu et al., 2022) and adopt lightweight task-specific adapters $\{\epsilon^t\}_{t=1}^T$ using a pre-trained CLIP backbone:

$$\mathcal{F}^t = \left\{ \{\mathbf{E}_{\text{text}}, \mathbf{E}_{\text{image}}\}, \{\epsilon^t\}_{t=1}^T \right\}. \tag{2}$$

Where $\epsilon^t(x_i)$ denotes expert t's prediction on $x_i$. To avoid catastrophic forgetting and reduce reliance on task-ID prediction during inference, we strictly isolate experts by freezing the shared backbone and all task-specific adapters except

the current one. Our design ensures that each expert learns independently without interference from other tasks. Unlike other parameter sharing methods, our approach encourages experts to maintain epistemic uncertainty outside their own domain rather than falsely generalizing to unseen tasks.

While expert isolation prevents interference between tasks, all experts still rely on the same pre-trained backbone for feature extraction. In our preliminary study on CLIP-based MoE continual learners, a counterintuitive yet interesting pattern consistently emerged: the model often appears confident, yet remains uncertain about task-relevant meaning. This is because its confidence reflects alignment strength between visual and text embeddings—not actual certainty about the label. For example, an image of a boy may be confidently matched with girl due to shared context in the pretraining corpus (Figure 1, (top)). These errors are not caused by epistemic uncertainty—the model has likely seen the feature—but rather by aleatoric uncertainty in the backbone's task-agnostic representation. Such shortcut features can activate multiple experts, making expert selection unreliable. However, past experts often have low epistemic uncertainty about these features, having encountered them in different task contexts. In the next section, we propose a temperature scaling strategy that leverages this asymmetry by leveraging predictive epistemic uncertainty from past experts to guide the training of new experts, helping resolve the ambiguity introduced by the shared backbone.

### 3.2. Temperature-Guided Expert Mixture

We propose a Temperature-Guided Expert Mixture mechanism based on the challenges discussed in Section 3.1. Confidence calibration via temperature scaling aims to align

a model's predicted probabilities with the true likelihood of correctness. Tu et al. (2024) show that the CLIP model suffers from poor calibration, often producing misleading confidence scores. Though the foundation model includes a learned temperature-like parameter $L$ (eq. 1) to scale contrastive similarity scores, this affects embedding sharpness and class separation rather than calibration quality.

To form a true expert mixture, we introduce temperature $\{\tau(\mathbf{x}_i^t)\}_{i=1}^{|D^t|}$ where each $\tau(\mathbf{x}_i^t)$ modifies an expert's logits. When $\tau(\mathbf{x}_i^t) > 1$, predictions become more uniform (lower confidence), and when $\tau(\mathbf{x}_i^t) < 1$, predictions become sharper (higher confidence). Traditional scaling improves **intra-expert** calibration but fails to enhance **inter-expert** mixture quality or OOD detection, as intra-expert scaling does not mitigate high-confidence OOD errors.

To improve the experts' awareness of OOD detection, we incorporate an adaptive threshold scaling mechanism. For this purpose, we first filter out samples from the current task dataset using the open-set error. The rationale behind this is that if previous experts, who were not trained on the current task's dataset, show high-confidence predictions on these samples, such high-confidence samples likely have a high semantic correlation with their corresponding training sets. Additionally, since experts are independent, their logits are not directly comparable. Therefore, we use distinct thresholds for each expert based on their overall confidence in the current task. This automated and adaptive mechanism ensures that only high-confidence samples by each expert are selected. For a previous task, $p < t$, the top-k confidence set in the eyes of an expert $\epsilon^p$ is:

$$\text{conf}^p = \underset{S \subset \mathcal{D}^t, \ |S|=k}{\arg\max} \sum_{\mathbf{x}^t \in S} \epsilon^p(\mathbf{x}^t), p = 1, \dots, t-1. \quad (3)$$

Finally, we apply temperature scaling to the filtered samples differently to adjust their confidence. This results in a clear distinction between in-distribution (IND) and OOD samples, as the model adjusts confidence on semantically similar samples. With pre-defined upper bound $\tau_{UB}$ and lower bound $\tau_{LB}$ we can rewrite equation 1 with the adaptive threshold temperature $\tau(\mathbf{x}_i^t)$ as follows:

$$\mathcal{L}^t = \frac{1}{|\mathcal{D}^t|} \sum_{i=1}^{|\mathcal{D}^t|} \frac{1}{\tau(\mathbf{x}_i^t)} \cdot \mathcal{L}_{\text{CE}}(\boldsymbol{\theta}^t; \mathbf{x}_i^t, y_i^t);$$
$$\text{where} \quad \tau(\mathbf{x}_i^t) = \begin{cases} \tau_{LB}, & \mathbf{x}_i^t \in \bigcup_{p=0}^{t-1} \text{conf}^p \\ \tau_{UB}, & \text{otherwise} \end{cases} \quad (4)$$

Although the proposed temperature-guided training is operationally local to each expert at present, it implicitly breaks the strict task boundary in continual learning problems. Without introducing additional memory or replay, each current expert is regularized by the statistical properties of the prior task captured through confidence-scaled gradients

from early experts. To better understand this effect, we formalize our scaling act as inducing an epistemic prior in the general Bayesian inference, and we show that the prior adaptively sharpens the posterior over the current expert in regions of shortcut activated samples. We show that this encourages the current expert to increase its predictive certainty over such samples, effectively amplifying a confidence gap that can later assist in expert selection at test time. This theoretical view supports our central claim: that temperature scaling not only improves local task fitting but also enhances global task coordination in continual learning.

**Proposition 3.1** (Equivalence of temperature scaling to general bayesian inference under epistemic energy prior). *The temperature-scaled objective $\frac{1}{\tau(\boldsymbol{x})} \mathcal{L}_{CE}(\boldsymbol{\theta}^t; \boldsymbol{x}, y)$ corresponds to a posterior update under Generalized Bayesian Inference (GBI) with an epistemic prior of Boltzmann distribution form: $p_E(\boldsymbol{\theta}^t) \propto \exp\{E(\boldsymbol{\theta}^t)\}$, where $E(\boldsymbol{\theta}^t)$ is a confidence-guided energy regularizer derived from a previous expert's belief.*

**Theorem 3.2** (Confidence separation over finite shortcut ambiguous samples via $\delta$-temperature scaling). *Specify each $\epsilon^t$ as a softmax classifier, whose posterior is approximated by $q_t(\boldsymbol{\theta})$ via black-box variational inference maximizing the ELBO of the GBI objective:*

$$\mathcal{F}[q_t] = KL[q_t || p_0] + \sum_{\boldsymbol{x}, y \in \mathcal{D}_t} a(\boldsymbol{x}) \mathbb{E}_{q_t}[\mathcal{L}(\boldsymbol{\theta}; \boldsymbol{x}, y)] \quad (5)$$

*where $a(\boldsymbol{x}) = \frac{1}{\tau(\delta)-1}$, with $\delta = conf^{t-1}(\boldsymbol{x}) := \max_y \epsilon^{t-1}(\boldsymbol{x})$ and $p_0$ is the prior. Define $\delta$-shortcut activated region: $A_{t-1}^\delta = \{\boldsymbol{x} \in \mathcal{D}^t | conf^{t-1}(\boldsymbol{x}) \geq \delta\}$. Then under the softmax likelihood and Laplace approximation to $q_t(\boldsymbol{\theta})$, the following confidence improvement holds for all $\boldsymbol{x} \in A_{t-1}^\delta$:*

$$\frac{1}{|A_{t-1}^\delta|} \sum_{\boldsymbol{x} \in A_{t-1}^\delta} [conf^t(\boldsymbol{x}) - conf^{t-1}(\boldsymbol{x})] \geq g(\delta) = C \cdot \tau(\delta) \sigma^2$$
$$(6)$$

*where $\sigma^2$ denotes the epistemic variance under $q_{t-1}(\boldsymbol{\theta})$ and $C > 0$ is a constant that depends on model curvature.*

**Proof sketch.** We extend the intuitive definition of shortcut features by Geirhos et al. (2020) and define a set $A_{t-1}^\delta$ of shortcut-activated samples in the context of incremental continual learning with MoE. During the posterior learning, our temperature scaling assigns higher ELBO weights to those samples, simulating their repetition and as a result, amplifies posterior curvature (i.e., Hessian of the log posterior) and shrinks the predictive variance (Kruschke, 2010; Bissiri et al., 2016). Under the Laplace approximation, the posterior mean remains stable, so predictive confidence increases with variance contraction. Then we can lower-bound confidence gain over $A_{t-1}^\delta$ as a function of $\delta$, $\tau$, and prior uncertainty. ∎

*Table 1.* Performance on **CIFAR-100** and **TinyImageNet** in the **CIL** setting. "Average" and "last" accuracies (%) are reported across incremental steps. Δ indicates gain over second-best.

| | CIFAR100 | | | | | | TinyImageNet | | | | | |
|---|---|---|---|---|---|---|---|---|---|---|---|---|
| | 10 step | | 20 step | | 50 step | | 5 step | | 10 step | | 20 step | |
| Method | Avg. ↑ | Last ↑ | Avg. ↑ | Last ↑ | Avg. ↑ | Last ↑ | Avg. ↑ | Last ↑ | Avg. ↑ | Last ↑ | Avg. ↑ | Last ↑ |
| UCIR (Hou et al., 2019) | 58.66 | 43.39 | 58.17 | 40.63 | 56.86 | 37.09 | 50.30 | 39.42 | 48.58 | 37.29 | 42.84 | 30.85 |
| DyTox (Douillard et al., 2022) | 74.10 | 62.34 | 71.62 | 57.43 | 68.90 | 51.09 | 55.58 | 47.23 | 52.26 | 42.79 | 46.18 | 36.21 |
| CLIP Zero-shot | 74.47 | 65.92 | 75.20 | 65.74 | 75.67 | 65.94 | 69.62 | 65.30 | 69.55 | 65.59 | 69.49 | 65.30 |
| Fine-tune | 65.46 | 53.23 | 59.69 | 43.13 | 39.23 | 18.89 | 61.54 | 46.66 | 57.05 | 41.54 | 54.62 | 44.55 |
| LwF (Li & Hoiem, 2017) | 65.86 | 48.04 | 60.64 | 40.56 | 47.69 | 32.90 | 60.97 | 48.77 | 57.60 | 44.00 | 54.79 | 42.26 |
| iCARL (Rebuffi et al., 2017) | 79.35 | 70.97 | 73.32 | 64.55 | 71.28 | 59.07 | 77.02 | 70.39 | 73.48 | 65.97 | 69.65 | 64.68 |
| LwF-VR (Ding et al., 2022) | 78.81 | 70.75 | 74.54 | 63.54 | 71.02 | 59.45 | 77.56 | 70.89 | 74.12 | 67.05 | 69.94 | 63.89 |
| ZSCL (Zheng et al., 2023) | 82.15 | 73.65 | 80.39 | 69.58 | 79.92 | 67.36 | 80.27 | 73.57 | 78.61 | 71.62 | 77.18 | 68.30 |
| MoE-Adapters (Yu et al., 2024) | 85.21 | 77.52 | 83.72 | 76.20 | 83.60 | 75.24 | 81.12 | 76.81 | 80.23 | 76.35 | 79.96 | 75.77 |
| RAIL (Xu et al., 2024) | 84.03 | 76.09 | 84.82 | 76.01 | 85.11 | 76.10 | 75.88 | 71.97 | 75.90 | 72.09 | 75.85 | 72.07 |
| **Ours** | **87.28** | **80.6** | **86.4** | **79.32** | **85.68** | **77.34** | **82.42** | **78.95** | **82.13** | **78.31** | **81.64** | **77.42** |
| Δ | +2.4% | +3.9% | +1.9% | +4.1% | +0.7% | +1.6% | +1.6% | +2.7% | +2.4% | +2.6% | +2.1% | +2.1% |

This sample-level confidence gain also hints at a broader effect. Since shortcut features remain easy to fit, achieving higher confidence under temperature scaling implicitly drives the expert to discover additional task-relevant cues and thus move beyond shortcut reliance without replay.

### 3.3. Distribution-Aware Weighting

While our adaptive threshold-based scaling improves OOD detection, we observe that the softmax of expert logits still yields overconfident posteriors for specific samples. Recent work has shown that Mahalanobis distance (MD) and Gaussian-based approximations are practical for OOD detection (Lee et al., 2018). However, these methods are often used in CL with memory buffers (Lin et al., 2024), which are impractical and unscalable in real-world scenarios. In contrast, multimodal model embeddings capture richer semantics than traditional deep networks, enabling samples to cluster meaningfully, even across classes. This property allows us to characterize in-distribution (IND) behavior in embedding space.

To incorporate MD-based weighting, we first embed the training set into a low-dimensional space using the task expert's visual encoder. We model the data distribution with a multivariate Gaussian. Specifically, let $\mathbf{v}_i^t \in \mathbb{R}^D$ denote the visual embedding of sample $i$ from the expert trained on task $t$, where $D$ is the hidden dimensionality of the encoder. We then estimate the empirical mean and covariance of the embeddings as: $\boldsymbol{\mu}^t = \frac{1}{|\mathcal{D}^t|} \sum_{i=1}^{|\mathcal{D}^t|} \mathbf{v}_i^t$, $\boldsymbol{\Sigma}^t = \frac{1}{|\mathcal{D}^t|-1} \sum_{i=1}^{|\mathcal{D}^t|} (\mathbf{v}_i^t - \boldsymbol{\mu}^t)(\mathbf{v}_i^t - \boldsymbol{\mu}^t)^T$. Where $\boldsymbol{\mu}^t \in \mathbb{R}^D$ and $\boldsymbol{\Sigma}^t \in \mathbb{R}^{D \times D}$. For a test sample $\mathbf{x}_{\text{test}}$, we obtain its embeddings from each task expert, denoted $\{\mathbf{v}_{\text{test}}^t\}_{t=1}^T$, and compute the MD to each task's distribution. A smaller MD indicates the sample is closer to that task's in-distribution (IND). The MD for a test sample $\mathbf{x}_{\text{test}}$ from task $t$ is:

$$\mathrm{MD}^t(\mathbf{v}_{\text{test}}^t) = \sqrt{(\mathbf{v}_{\text{test}}^t - \boldsymbol{\mu}^t)^T \boldsymbol{\Sigma}^{t-1} (\mathbf{v}_{\text{test}}^t - \boldsymbol{\mu}^t)} \quad (7)$$

We apply softmax to convert inverse MDs into a normalized weight distribution. We use a small value $\zeta$ in softmax to make the weight distribution more selective. Finally, we apply weights to each expert's sample-specific probability distribution over the classes. This results in test sample-specific weights for each expert, rather than assigning fixed weights per expert. The final prediction is:

$$\hat{y}_{\text{test}} = \arg\max \mathrm{concat}_t \left[ w^t(\mathbf{x}_{\text{test}}) \cdot \epsilon^t(\mathbf{x}_{\text{test}})[:] \right] ;$$
$$w^t(\mathbf{x}_{\text{test}}) = \mathrm{softmax}\left( \left[ \mathrm{MD}^t(\mathbf{v}_{\text{test}}^t) \times \zeta \right]^{-1} \right) \quad (8)$$

Here, [:] denotes slicing and $\hat{y}_{\text{test}}$ is the prediction for $\mathbf{x}_{\text{test}}$. Figure 2 illustrates our framework.

## 4. Experiments

**Evaluation overview.** We design our experimental setting based on recent literature. Our main experiment is divided into three parts. First, we conduct experiments on the traditional CIL benchmark. Then, we conduct experiments on the more challenging subpopulation shift benchmarks. Finally, we conduct experiments on uneven subpopulation shift to further demonstrate the robustness of the model. We also perform experiments on the X-TAIL benchmark. Additionally, we conduct ablation studies to justify the effectiveness of different proposed components in our method. Implementation details are available in Appendix I.

**Baseline.** We compare our method with eight CLIP-backbone-based baselines: CLIP Zero-shot, Fine-tune, LwF (Li & Hoiem, 2017), iCARL (Rebuffi et al., 2017), LwF-VR (Ding et al., 2022), ZSCL (Zheng et al., 2023), MoE-Adapters (Yu et al., 2024), RAIL (Xu et al., 2024), and additional non CLIP backbone based baselines: UCIR (Hou et al., 2019) and DyTox (Douillard et al., 2022). For subpopulation shift benchmarks, we compare our method with four CLIP-backbone-based baselines. For ZSCL, we use their reported reference dataset, Conceptual Caption (CC) (Sharma

*Table 2.* Performance comparison in **sub-population shift incremental learning** setting on **Entity-13** (5 steps) and **Entity-30** (4 steps) benchmarks. We report "**seen**", "**unseen**" (in parentheses), and "**All**" accuracies (%) across steps. $\Delta$ indicates gain over second-best.

| Steps | 0 | 1 | 2 | 3 | 4 | 5 | All |
|---|---|---|---|---|---|---|---|
| Method | \multicolumn{7}{c}{Breeds Entity 13 Benchmark with 5 Steps (Even Update)} | | | | | | |
| CLIP Zero-shot | 75.09 | 71.2 | 70.10 | 70.77 | 72.17 | 72.28 | 71.93 |
| ZSCL (Zheng et al., 2023) | 69.86 (66.10) | 86.56 (73.17) | 86.19 (77.84) | 86.27 (74.69) | 85.78 (83.92) | 83.52 | 79.62 |
| MoE-Adapters (Yu et al., 2024) | 97.08 (79.72) | 92.54 (68.86) | 87.26 (69.05) | 86.97 (70.15) | 84.74 (78.23) | 74.06 | 79.43 |
| RAIL (Xu et al., 2024) | 96.30 (71.30) | 89.98 (71.33) | 88.49 (71.74) | 87.29 (71.74) | 87.29 (72.28) | 85.95 | 81.06 |
| **Ours** | **97.09 (79.75)** | **96.53 (79.53)** | **93.65 (81.25)** | **91.7 (80.84)** | **90.36 (84.46)** | **89.71** | **87.08** $\Delta$ **+7.43%** |
| | \multicolumn{7}{c}{Breeds Entity 30 Benchmark with 4 Steps (Even Update)} | | | | | | |
| CLIP Zero-shot | 68.53 | 65.3 | 65.64 | 66.28 | 66.03 | - | 66.35 |
| ZSCL (Zheng et al., 2023) | 56.82 (55.93) | 84.32 (76.35) | 86.18 (73.37) | 85.93 (74.8) | 84.41 | - | 76.96 |
| MoE-Adapters (Yu et al., 2024) | 95.92 (**72.42**) | 88.23 (70.13) | 83.28 (65.1) | 81.63 (70.4) | 75.39 | - | 77.06 |
| RAIL (Xu et al., 2024) | 95.50 (65.81) | 89.22 (65.98) | **87.01 (66.15)** | 86.30 (66.03) | 85.25 | - | 78.63 |
| **Ours** | **96.0 (72.13)** | **94.69 (77.80)** | 71.38 (**79.5**) | **90.22 (76.33)** | **88.84** | - | **84.88** $\Delta$ **+7.95%** |

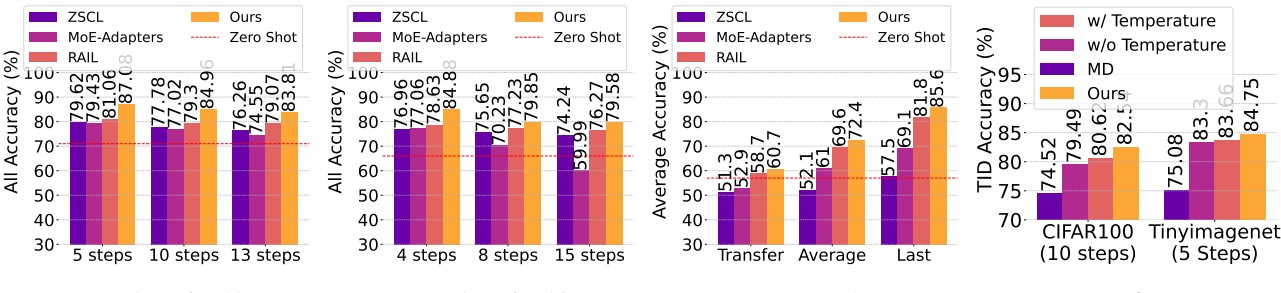

*Figure 3.* Entity-13.  *Figure 4.* Entity-30.  *Figure 5.* X-TAIL.  *Figure 6.* TID.

et al., 2018). For MoE-Adapters, we match the number of experts with ours to ensure a fair comparison. For RAIL, we use the primal form. We use other settings and hyper-parameters as reported in the respective works. We utilize publicly available results when possible.

### 4.1. Class-Incremental Learning Performance

**Setting.** Following Zheng et al. (2023); Yu et al. (2024), to evaluate in the traditional CIL setting, we use the CIFAR-100 (Krizhevsky, 2009) and TinyImageNet (Le & Yang, 2015) datasets. For CIFAR-100, we divide the 100 classes into 10, 20, and 50 steps. For TinyImageNet, which has 200 classes, we initially learn 100 classes, with the remaining classes learned in increments of 5, 10, and 20 steps. For example, if a dataset has a total of M classes and we divide it into K steps, then each step introduces M/K new classes. We report two metrics: **Average**, the mean accuracy across all tasks and timestamps, and **Last**, the mean accuracy after the final task. See Appendix C.1 for more.

**Analysis.** Table 1 shows the CIL results. Our method consistently achieves the highest "Average" and "Last" accuracies, outperforming all competing methods, including recent baselines RAIL and MoE-Adapters. RAIL uses single modality (vision) and `memory buffer`, while MoE-Adapters employs trainable router for task ID prediction. The shared experts in MoE-Adapters are prone to forgetting and task detection errors; in contrast, our method mitigates

both by compartmentalizing experts. Although task accuracy depends on OOD detection, our adaptive temperature scaling, guided by Mahalanobis distance using multivariate Gaussian distributions address this challenge.

### 4.2. Sub-population Shift Incremental Performance

**Setting.** For the subpopulation shift evaluation, we use the BREEDS benchmark (Santurkar et al., 2021), which simulates real-world subpopulation shifts based on the ImageNet dataset (Deng et al., 2009). Specifically, we utilize the Entity-13 and Entity-30 benchmarks, which contain 13 and 30 superclasses and 260 and 240 subclasses, respectively. We follow the protocol proposed by Liang et al. (2022). For Entity-13, we use 5 incremental steps, while for Entity-30, we use 4 steps. In Entity-13, the first task includes 10 subclasses per superclass, with 130 unseen subclasses for incremental updates; in the 5-step setup, each task adds 2 new subclasses per superclass. Similarly, in Entity-30, the initial task includes 4 subclasses per superclass, with 120 unseen subclasses for incremental updates; in the 4-step setup, each step adds 1 new subclass per superclass. Both benchmarks simulate an even update, meaning that during each incremental step, there will be samples for each superclass. Subpopulations are strictly unseen and disjoint for each increment. In both protocols, training, and inference are performed using superclass labels, while samples are introduced with subclass labels. We report **Seen**, **Unseen**,

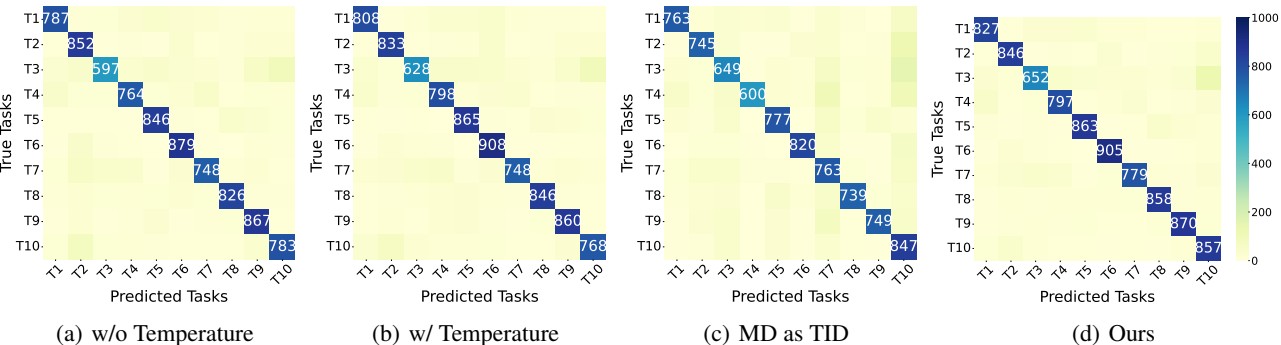

Figure 7. Comparison of different components of our method on the CIFAR-100 with 10 steps.

and **All** accuracy, averaged incremental accuracy on seen tasks, unseen tasks, and both, respectively.

**Analysis.** Table 2 shows results on Entity-13 and Entity-30 with evenly introduced subpopulation shifts. The main challenge is adapting to new subclasses without forgetting previously learned ones. For example, if the model initially learns to classify "goose" as a type of "bird" and then encounters a shift to the "flamingo" subpopulation, it should learn to classify "flamingo" as "bird" without forgetting how to classify "goose" as "bird". Our method consistently outperforms baselines on both benchmarks, maintaining balanced performance across seen and unseen tasks throughout all steps. In contrast, ZSCL initially lags due to parameter regularization but improves over time; MoE-Adapters and RAIL perform well early but struggle to retain seen knowledge. Overall, our method achieves stable, balanced performance. See Appendix F for details.

### 4.3. Models Robustness

**Setting.** We conduct an analysis of the robustness of our method in a more challenging subpopulation shift setting with uneven updates. The uneven updates mimic the situation where part of the population (i.e., superclasses) receives samples (i.e., from unseen subclasses) during training. We also use a longer step size to measure its ability to stay unbiased toward newer tasks. We use the Entity-13 and Entity-30 benchmarks with 10 and 13 steps for the former, and 8 and 15 steps for the latter, respectively. As the model receives unbalanced and uneven updates, learning new classes and preserving knowledge of old classes, and long-run bias are additional key issues with subpopulation shifts. We follow Xu et al. (2024) to present results on the X-TAIL benchmark.

**Analysis.** Figures 3 and 4 present a performance comparison on the Entity-13 and Entity-30 benchmarks, using both "even" (5 steps for Entity-13, 4 steps for Entity-30) and "uneven" (10 and 13 steps for Entity-13, 8 and 15 steps for Entity-30) update intervals. This is a more realistic setting

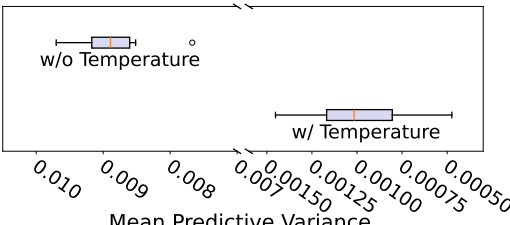

Figure 8. Effect of $\delta$-temperature scaling from Theorem 3.2.

than the benchmark described in section 4.2, as samples from all subpopulations might not always be available for training. For example, when the model is learning about "garment" and "insect" categories, samples from both subpopulations, such as "T-shirt" and "butterfly" may arrive at the same time. However, at another timestamp, only a sample from "garment" (e.g., "skirt") might arrive, with no new samples from "insect". This situation can arise for various reasons, such as data unavailability, differences in subpopulation sizes (i.e., not all populations have the same number of subpopulations), or data imbalance. In Figures 3 and 4, we observe that our method outperforms all baselines on the "All" metric, demonstrating balanced learning between seen and unseen sub-populations in even and uneven update intervals, as well as in both short- and long-term scenarios. Figure 5 shows our model's robustness on X-TAIL versus three baselines (details on Appendix G).

**Effect of $\delta$-temperature scaling from Theorem 3.2.** We analyze predictive variance distributions across CIFAR100 (10 steps) with and without temperature scaling (Figure 8). Without scaling, the distribution shows a heavy high-variance tail; with scaling, both the mean and tail contract, yielding a more concentrated, near-Gaussian shape. This non-uniform contraction indicates selective variance reduction, consistent with our theory: the temperature-induced epistemic prior increases curvature along shortcut-prone directions, accelerating epistemic convergence for shortcut-like samples (see Appendix B).

## 4.4. Ablation Study

We analyze the effects of temperature scaling and MD on task ID detection. Figure 7 shows heatmaps of task ID detection on CIFAR100 after learning all tasks. Independently trained experts show clear task detection performance (Figure 7(a)). However, our simple yet effective adaptive threshold temperature scaling improves task separation by reducing overlap between task-specific outputs (Figure 7(b)). While the diagonal elements clearly demonstrate that most tasks benefit from scaling, a few tasks are also impacted by side effects. The multivariate Gaussian is also effective for the task ID detection using MD as a weight (Figure 7(c)). Finally, our combined method, which benefits from both temperature scaling and MD, shows the best overall performance (Figure 7(d)). Figure 6 shows the overall task ID prediction accuracy on two datasets. Overall, our method yields more diverse and generalizable predictions. Additional ablations are provided in Appendix D.

## 5. Conclusions

We introduce a multimodal approach for continual learning, addressing diverse settings like class-incremental, domain-incremental, and subpopulation-incremental learning. Our analysis identifies key challenges, such as reliance on short-cut features and the need for sample task ID prediction, which limit the adaptability of the multimodal model. To address these, we propose adaptive threshold scaling and a distribution-aware expert weighting mechanism, enabling dynamic task adjustment and reducing spurious correlations. We provide a theoretical view to support our claim. Our method is reference dataset-free and memory-free, making it efficient. It demonstrates robust performance and significant improvements over baselines across diverse benchmarks.

## Software and Data

We include a link to our source code in the Appendix J to ensure reproducibility. We use publicly available datasets in all of our experiments.

## Acknowledgements

We thank the anonymous reviewers for their constructive feedback and helpful suggestions.

## Impact Statement

This paper presents work whose goal is to advance the field of Machine Learning. There are many potential societal consequences of our work, none which we feel must be specifically highlighted here.

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

## A. Organization

We organize the Appendix into multiple sections. In the second section, we provide the proof of the theorem. The third section provides further details on class-incremental learning, including results on additional baselines and benchmarks, as well as qualitative and additional task ID detection results. In the fourth section, we provide additional ablation studies. In the fifth section, we present the impact of multimodality on learning. In the sixth section, we present further details and stepwise performance on sub-population shift incremental learning. Next, we provide detailed results on cross-domain task-agnostic incremental learning across 11 datasets. Finally, we discuss some limitations and implementation details, and provide instructions to access the source code to ensure the reproducibility of our method.

## B. Proofs of Theorem and Proposition

### B.1. Proof of Proposition 3.1

Let $\epsilon^t(\mathbf{x}) = p(y|\mathbf{x}) = \int p(y|\mathbf{x}, \boldsymbol{\theta}) p(\boldsymbol{\theta}|\mathcal{D}^t)$ denote the predictive distribution from a Bayes learner trained on the dataset $\mathcal{D}^t$ (*i.e.* expert t's prediction), where $\boldsymbol{\theta}$ denotes the tunable parameters. We want to formalize the process of learning $\boldsymbol{\theta}'$ for expert $t'$, $t' > t$, as general Bayes inference (Bissiri et al., 2016) under epistemic guided prior.

A typical posterior in GBI is defined as

$$p(\boldsymbol{\theta}|\mathcal{D}) \propto p(\boldsymbol{\theta}) \exp\left\{-\sum_i \eta_i \mathcal{L}(\boldsymbol{\theta}; \mathbf{x}_i, y_i)\right\} \tag{9}$$

where $p(\boldsymbol{\theta})$ is the prior distribution of $\boldsymbol{\theta}$. $\mathcal{L}_i$ is the per-sample loss function. $\eta_i$ is the trust/precision parameter.

We first define a joint prior $p_{\text{Joint}}(\boldsymbol{\theta}')$ over $\boldsymbol{\theta}'$ with **two components**.

1. **A base prior** $p_0(\boldsymbol{\theta}')$ shared by all experts. (*i.e.* initialization belief)

2. **An epistemic prior** $p_{\text{E}}(\boldsymbol{\theta}'|\epsilon^t)$.

Thus,

$$p_{\text{Joint}}(\boldsymbol{\theta}') := p_0(\boldsymbol{\theta}') p_{\text{E}}(\boldsymbol{\theta}'|\epsilon^t) \tag{10}$$

We further assume that the influences of different data samples are conditionally independent given a former expert's prediction. Such that the following factorization holds:

$$p_{\text{E}}(\boldsymbol{\theta}') = \frac{1}{Z_{\text{E}}} \prod_i^{|\mathcal{D}^{t'}|} \psi_i(\boldsymbol{\theta}'; \epsilon^t(\mathbf{x}_i)) \tag{11}$$

where $Z_{\text{E}}$ is the normalization constant.

Substituting (10) back to (9) we have the posterior belief update given by

$$p(\boldsymbol{\theta}'|\mathcal{D}^{t'}) \propto p_0(\boldsymbol{\theta}') \prod_i^{|\mathcal{D}^{t'}|} \psi_i(\boldsymbol{\theta}'; \epsilon^t(\mathbf{x}_i)) \exp\left\{-\sum_{i=1}^{|D^{t'}|} \mathcal{L}(\boldsymbol{\theta}; \mathbf{x}_i, y_i)\right\}$$

$$= p_0(\boldsymbol{\theta}') \exp\left\{-\sum_{i=1}^{|\mathcal{D}^{t'}|} \mathcal{L}(\boldsymbol{\theta}'; \mathbf{x}_i, y_i) + \tilde{\mathcal{L}}_i(\boldsymbol{\theta}')\right\} \tag{12}$$

where $\tilde{\mathcal{L}}_i(\boldsymbol{\theta}') := -\ln \psi_i(\boldsymbol{\theta}'; \epsilon^t(\mathbf{x}_i))$. We now assume that the epistemic prior contributes an additive loss-like connection to posterior update, given by $\tilde{\mathcal{L}}_i(\boldsymbol{\theta}') = a_i \mathcal{L}(\boldsymbol{\theta}'; \mathbf{x}_i, y_i)$, then we can rewrite the GBI loss in (12) as $\mathcal{L}_{\text{eff}} = (1 + a_i)\mathcal{L}(\boldsymbol{\theta}'; \mathbf{x}_i, y_i)$, which means that the prior increases or decreases the effective strength of each data sample. We can further justify this structure from an energy-based perspective. $\psi_i(\boldsymbol{\theta}') = \exp\{E_i(\boldsymbol{\theta}')\}$ with $E_i(\boldsymbol{\theta}') := a_i \mathcal{L}(\boldsymbol{\theta}'; \mathbf{x}_i, y_i)$. This demonstrates the epistemic trust induced by the past expert: a small $a_i$ means a stronger prior, therefore suppresses the posterior update with sample $\mathbf{x}_i$, while a larger $a_i$ encourages the model to fit to $\mathbf{x}_i$.

Now let $\tau(\mathbf{x}_i)$ be a summarized confidence function of $\epsilon^t$,

$$\tau(\mathbf{x}_i) = h(\epsilon^t(\mathbf{x}_i)), \quad h := P_y \to (0, +\infty) \tag{13}$$

Setting $a_i = \frac{1}{\tau(\mathbf{x})} - 1$, the effective posterior update becomes

$$p(\boldsymbol{\theta}'|\mathcal{D}^{t'}) \propto p_0(\boldsymbol{\theta}') \prod_{i=1}^{|\mathcal{D}^{t'}|} p(y_i|\mathbf{x}_i; \boldsymbol{\theta}')^{\frac{1}{\tau(\mathbf{x}_i)}} \tag{14}$$

which is exactly the proposed confidence-guided temperature scaling learning objective.

*Remark* B.1. While the generalized Bayesian framework grants flexibility in the choice of loss functions, its true expressive power lies in how it emphasizes the interpretability and structure of the prior. In our formulation, we leverage this freedom to specify a data-dependent, epistemically grounded prior, which enables our confidence-based temperature scaling to emerge as a principled posterior update. This derivation shows that temperature scaling emerges as a special case of generalized Bayesian inference, where confidence-adjusted energy terms serve as epistemic priors regulating posterior updates.

*Remark* B.2. Note that each energy term $E(\boldsymbol{\theta}')$ in the epistemic prior is negative, so the epistemic prior is indeed in Boltzmann form, where lower loss implies higher prior belief.

## B.2. Proof of Theorem 3.2

We aim to quantify how scaling the ELBO with confidence-aware temperature improves predictive certainty in regions dominated by shortcut features.

Let $q_t(\boldsymbol{\theta}) = \mathcal{N}(\boldsymbol{\mu}_t, \boldsymbol{\Sigma}_t)$ be the Laplace approximation of the variational posterior for expert $t$.

Let $A_{t-1}^\delta := \{\mathbf{x} \in \mathcal{D}^t \mid \mathrm{conf}^{t-1}(\mathbf{x}) \geq \delta\}$ be the shortcut activated region induced by expert $t-1$.

Let $f_{\boldsymbol{\theta}}(\mathbf{x})$ denote the softmax output of sample $\mathbf{x}$.

Using the second-order delta method, the expected prediction is given by:

$$\mathbb{E}_{q_t}[f(\boldsymbol{\theta})] \approx f_{\boldsymbol{\mu}_t}(\mathbf{x}) + \frac{1}{2}\nabla_{\boldsymbol{\theta}} f_{\boldsymbol{\theta}}(\mathbf{x})^\top|_{\boldsymbol{\mu}_t} \cdot \boldsymbol{\Sigma}_t \cdot \nabla_{\boldsymbol{\theta}} f_{\boldsymbol{\theta}}(\mathbf{x})|_{\boldsymbol{\mu}_t} \tag{15}$$

Let $\mathbf{v}_x$ denote $\nabla_{\boldsymbol{\theta}} f_{\boldsymbol{\theta}}(\mathbf{x})|_{\boldsymbol{\mu}_t}$.

Then

$$\mathrm{conf}^t(\mathbf{x}) = \max_y \mathbb{E}_{q_t}[f_{\boldsymbol{\theta}}] \geq f_{\boldsymbol{\mu}_t}(\mathbf{x}) + \mathbf{v}_x^\top \boldsymbol{\Sigma}_t \mathbf{v}_x$$

Assume $\boldsymbol{\mu}_t \approx \boldsymbol{\mu}_{t-1}$ (the characteristic of shortcut feature),

$$\mathrm{conf}^{t-1}(\mathbf{x}) \approx f_{\boldsymbol{\mu}_t}(\mathbf{x})$$

Then

$$\mathrm{conf}^t(\mathbf{x}) - \mathrm{conf}^{t-1}(\mathbf{x}) \geq \mathbf{v}_x^\top \boldsymbol{\Sigma}_t \mathbf{v}_x$$

Since $q_t$ is a Laplace approximation,

$$\boldsymbol{\Sigma}_t = \left(\nabla_{\boldsymbol{\theta}}^2 \left[-\log q_t(\boldsymbol{\theta})\right]\right)^{-1}$$

From the GBI objective:

$$\mathcal{F}(q_t) = \mathrm{KL}(q_t \| p_0) + \sum_{\mathbf{x} \in \mathcal{D}_t} a(\mathbf{x})\mathbb{E}_{q_t}[\mathcal{L}(\boldsymbol{\theta}, \mathbf{x}, y]$$

Take the Hessian on both sides and rearrange:

$$\nabla_{\boldsymbol{\theta}}^2[-\ln q_t(\boldsymbol{\theta})] = \nabla_{\boldsymbol{\theta}}^2[-\ln p_0] + \sum a(\mathbf{x})\nabla_{\boldsymbol{\theta}}^2[-\ln p(y \mid \mathbf{x}, \boldsymbol{\theta})]$$

So when a sample receives weight $a(x) = \frac{1}{\tau(\delta)-1}$, we have

$$\boldsymbol{\Sigma}_t = \tau(\delta) \cdot H_0, \quad \text{where } H_0 = \nabla^2_\theta[-\ln q_t(\boldsymbol{\theta})] \text{ with } \tau(\delta) \geq 1 \text{ (i.e., } \mathbf{x} \notin A^\delta_{t-1})$$

Which implies that

$$\text{Var}_{q_t}[f_{\boldsymbol{\theta}}(\mathbf{x})] = \mathbf{v}_x^\top \boldsymbol{\Sigma}_t \mathbf{v}_x = \tau(\delta)\mathbf{v}_x^\top H_0^{-1}\mathbf{v}_x = \tau(\delta) \cdot \sigma^2$$

where $\sigma^2$ denotes the variance in the direction $V_x$ under unscaled posterior gradient.

Thus, we have:

$$\text{conf}^t(\mathbf{x}) - \text{conf}^{t-1}(\mathbf{x}) \geq \frac{1}{2}\tau(\delta)\|\mathbf{v}_x\|^2 \cdot \sigma^2$$

Averaging over $\mathbf{x} \in A^\delta_{t-1}$, and denote $C := \frac{1}{|A^\delta_{t-1}|}\sum_{\mathbf{x} \in A^\delta_{t-1}}\|\mathbf{v}_x\|^2$,

We have:

$$\frac{1}{|A^\delta_{t-1}|}\sum_{\mathbf{x} \in A^\delta_{t-1}}(\text{conf}^t(\mathbf{x}) - \text{conf}^{t-1}(\mathbf{x})) \geq \frac{1}{2}\tau(\delta) \cdot C \cdot \sigma^2 \quad \blacksquare$$

**B.3. Empirical analysis of $\delta$-temperature scaling from Theorem 3.2.**

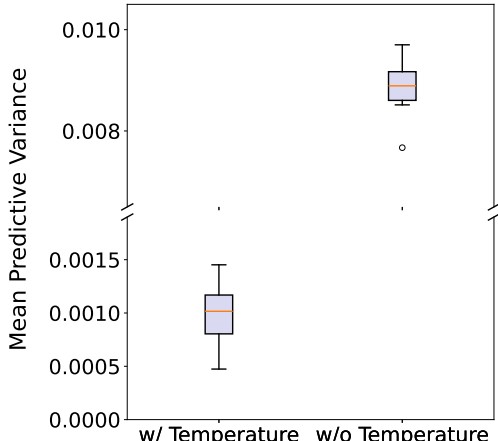

*Figure 9.* Effectiveness of $\delta$-temperature scaling from Theorem 3.2.

We analyze the distribution of predictive variances across all continual learning tasks, with and without our temperature scaling-based epistemic prior (See Figure 9) on the CIFAR100 dataset with 10 steps. While we cannot directly measure shortcut samples due to the lack of ground-truth annotation, the distributional behavior itself is informative. If temperature scaling had a uniform influence on all samples, the predictive variances would decrease at roughly the same rate, resulting in a simple downward shift of the distribution while preserving its shape. However, this is not what we observe. The baseline (w/o temperature) exhibits a pronounced long-tail of high-variance samples, whereas our method (w/ temperature) not only lowers the mean variance but also substantially contracts the tail, producing a more concentrated, approximately Gaussian-like distribution. This change in distributional shape indicates that variance reduction is not uniform across samples. Instead, a specific subset experiences disproportionately faster epistemic variance contraction. This selective contraction is precisely the behavior predicted by our theoretical analysis: the epistemic prior introduced by temperature scaling imposes stronger curvature along shortcut-prone directions, causing posterior variances associated with those regions to converge more rapidly. Thus, even without explicit shortcut labels, the observed collapse of the high-variance tail provides indirect but consistent empirical evidence supporting our theoretical claim that temperature scaling accelerates epistemic convergence for shortcut-like samples.

# C. Class Incremental Learning

## C.1. Additional CIL Baselines and Benchmark

**Setting.** We follow the same setting as Section 4 to report results on additional class incremental learning baselines. Specifically, we focus on recent methods built on the CLIP (Radford et al., 2021) backbone, as is our approach, and designed for the CIL setting. The baselines are CLAP4CLIP (Jha et al., 2024), CIL-CLIP (Huang et al., 2024), and PROOF (Zhou et al., 2025). We evaluate on CIFAR-100 (B0 Inc10), where each task consists of 10 classes. In addition, we compare with ImageNet-R (Hendrycks et al., 2021), where each task contains 20 incremental classes (B0 Inc20). The notation (B-x Inc-y) denotes the split, where Base-x indicates the number of classes in the first stage, and Inc-y indicates the number of new classes in each subsequent task. Here, x = 0 means that each task contains y classes (Zhou et al., 2025). We follow Huang et al. (2024) to design ImageNet-R (B0 Inc20).

**Analysis.** Table 3 reports the comparative results. Overall, our method consistently outperforms prior approaches on both CIFAR-100 (B0 Inc10) and ImageNet-R (B0 Inc20). Notably, despite not relying on any replay buffer, our approach achieves the highest average and last-task accuracies across both benchmarks. This highlights our methods effectiveness and efficiency compared to existing methods.

*Table 3.* Results on class-incremental learning with CIFAR-100 (B0 Inc10) and ImageNet-R (B0 Inc20). Average (Avg) and Last accuracies (%) are reported across incremental steps.

| Method | CIFAR100 (B0 Inc10) | | ImageNet-R (B0 Inc20) | | Exemplar |
|---|---|---|---|---|---|
| | Avg | Last | Avg | Last | |
| CLAP4CLIP (Jha et al., 2024) | 86.13 | 78.21 | 85.77 | 79.98 | Required |
| CIL-CLIP (Huang et al., 2024) | 86.19 | 79.04 | 85.58 | 80.28 | Required simulated replay |
| PROOF (Zhou et al., 2025) | 86.70 | 79.05 | 85.34 | 80.10 | Required |
| Ours | **87.28** | **80.6** | **86.91** | **80.95** | Not Required |

*Table 4.* Results (Last Accuracy) with standard deviations over three runs.

| Method | CIFAR100 | | | TinyImageNet | | |
|---|---|---|---|---|---|---|
| | 10 step | 20 step | 50 step | 5 step | 10 step | 20 step |
| MoE-Adapters | 77.60 ± 0.21 | 76.12 ± 0.18 | 74.69 ± 0.78 | 76.97 ± 0.23 | 76.11 ± 0.33 | 75.94 ± 0.23 |
| RAIL | 75.87 ± 0.31 | 76.36 ± 0.49 | 76.03 ± 0.39 | 71.68 ± 0.41 | 71.89 ± 0.28 | 72.10 ± 0.15 |
| Ours | 80.70 ± 0.26 | 79.29 ± 0.22 | 77.20 ± 0.48 | 78.90 ± 0.25 | 78.35 ± 0.31 | 77.48 ± 0.18 |

## C.2. Additional Details for Table 1 (Main Paper)

As described in Section 4 of the main paper, we rely on publicly available results. To provide standard deviation information, we provide additional details for Table 1 from the main paper in Table 4.

*Table 5.* Expert IDs and corresponding In-Distribution (IND) Classes for the CIFAR100 Dataset with 10 Steps. Examples included in both the analysis and Figure 10 are **bold-underlined**, while those only presented in Figure 10 are underlined.

| Expert ID | In-distribution Classes |
|---|---|
| Expert 1 | television, apple, oak_tree, pickup_truck, lizard, trout, road, wolf, mushroom, camel |
| Expert 2 | wardrobe, tulip, bowl, seal, mountain, snake, plate, butterfly, bicycle, telephone |
| Expert 3 | table, willow_tree, caterpillar, cockroach, flatfish, lobster, tiger, boy, beaver, ray |
| Expert 4 | rocket, raccoon, snail, maple_tree, spider, turtle, dinosaur, mouse, pear, sweet_pepper |
| Expert 5 | castle, streetcar, lawn_mower, bridge, house, pine_tree, couch, chair, squirrel, shark |
| **Expert 6** | aquarium_fish, cup, bee, man, **poppy**, sunflower, orange, bottle, elephant, skunk |
| Expert 7 | kangaroo, porcupine, forest, shrew, crocodile, clock, hamster, bear, can, chimpanzee |
| **Expert 8** | plain, cattle, **rose**, train, tractor, lion, bed, leopard, rabbit, skyscraper |
| Expert 9 | lamp, dolphin, cloud, tank, baby, whale, palm_tree, motorcycle, sea, possum |
| Expert 10 | woman, bus, worm, beetle, fox, otter, orchid, crab, girl, keyboard |

## C.3. Qualitative Results

**Setting.** In Figure 10, we present the qualitative results of our method on the CIFAR100 dataset with 10 steps. **True** denotes the true label, **Predicted** represents the predicted label with the predicted probability (inside parentheses), and **Second** indicates the second-highest probability after the predicted one. The top group of images (10(a)) represents class-incremental learning predictions by independent experts without any scaling applied during training. The middle group (10(b)) shows predictions made by our method on the same test images as the first group. The last group (10(c)) displays training images selected by our method during training for scaling. We present Table 5, which shows the Expert IDs and their corresponding in-distribution (IND) classes to provide a clearer illustration of the analysis. Additionally, we highlight the Expert IDs and the relevant classes discussed in the following analysis section.

**Analysis.** In Figure 10, we observe that, since no task-ID (TID) is provided during testing, multiple experts predict with high confidence. For example, in the **rose** image on the second row rightmost (10(a)), Expert No. 8, trained on "`rose`", makes the correct prediction. However, Expert No. 6, trained on "`poppy`", also makes a high-confidence prediction, resulting in both an incorrect TID prediction and a wrong final class prediction. In contrast, our method incorporates all previous experts, including Expert No. 6, during the training of Expert No. 8. This enables our method to identify samples contributing to the "open set error." In this case, specific "`rose`" images (such as the one in the 10(c) bottom row rightmost) from the task 8 train set, where Expert No. 6 made an open set error, are selected for temperature scaling. This allows the model to better account for the confidence levels of other experts. As a result, our method makes the correct prediction, as demonstrated in the **rose** image from 10(b) (second row rightmost). Some of the test images shown in the figure are even challenging for humans to classify. However, our dual task ID prediction mechanism, combining adaptive threshold scaling (which accounts for epistemic uncertainty, i.e., model uncertainty) and multivariate Gaussian Mahalanobis Distance (MD) (which accounts for aleatoric uncertainty, i.e., data uncertainty), effectively solves the TID detection problem.

This example highlights the advantages of introducing structured mechanisms over merely increasing the volume of memory space in continual learning. In real-world scenarios, the knowledge required to solve a task often spans multiple domains. We observe that, in many tasks, multiple experts provide rational predictions, which, in a sense, transcend the original problem definition to produce more accurate and knowledge-rich predictions. The effective collaboration mechanism among experts that we propose allows the MoE structure to demonstrate more organized and fine-grained control over memory retrieval. This capability enables the MoE to excel in handling complex, dynamic tasks, a feat that is challenging to achieve in traditional continual learning frameworks based on centralized, unstructured memory.

# D. Additional Ablation

## D.1. Ablation Study on Different Model Components

We conduct an additional ablation study on different components of our method using the TinyImageNet (Le & Yang, 2015) dataset with 5 steps. The base step contains a total of 100 classes, while each subsequent step includes 20 classes. Figure 11(a) presents the results on zero-shot CLIP, where we observe clear task separation demonstrating CLIP's OOD capabilities. Incorporating the MoE structure further improves performance, as shown in Figure 11(b). Adding temperature scaling enhances task separation further, though with minor side effects, as depicted in Figure 11(c). In Figure 11(d), we observe that the Mahalanobis distance (MD) from the learned embeddings' Gaussian distribution also reveals a clear pattern, though it achieves the lowest performance among the components. Finally, Figure 11(e) demonstrates the best overall performance by combining all components. There is a bias towards Task One due to the class imbalance across the other tasks.

## D.2. Ablation Study on Scalability

We conduct a scalability analysis concerning the number of model parameters. Our approach introduces approximately 1.9M trainable parameters per expert, constituting the only task-specific trainable component. This design ensures that most model parameters remain frozen, leading to computational efficiency and modular scalability. Table 6 presents a comparison against other CLIP-backbone-based baselines. For a fair comparison, we align the number of experts in our method with those used in the Mixture-of-Experts (MoE) baselines.

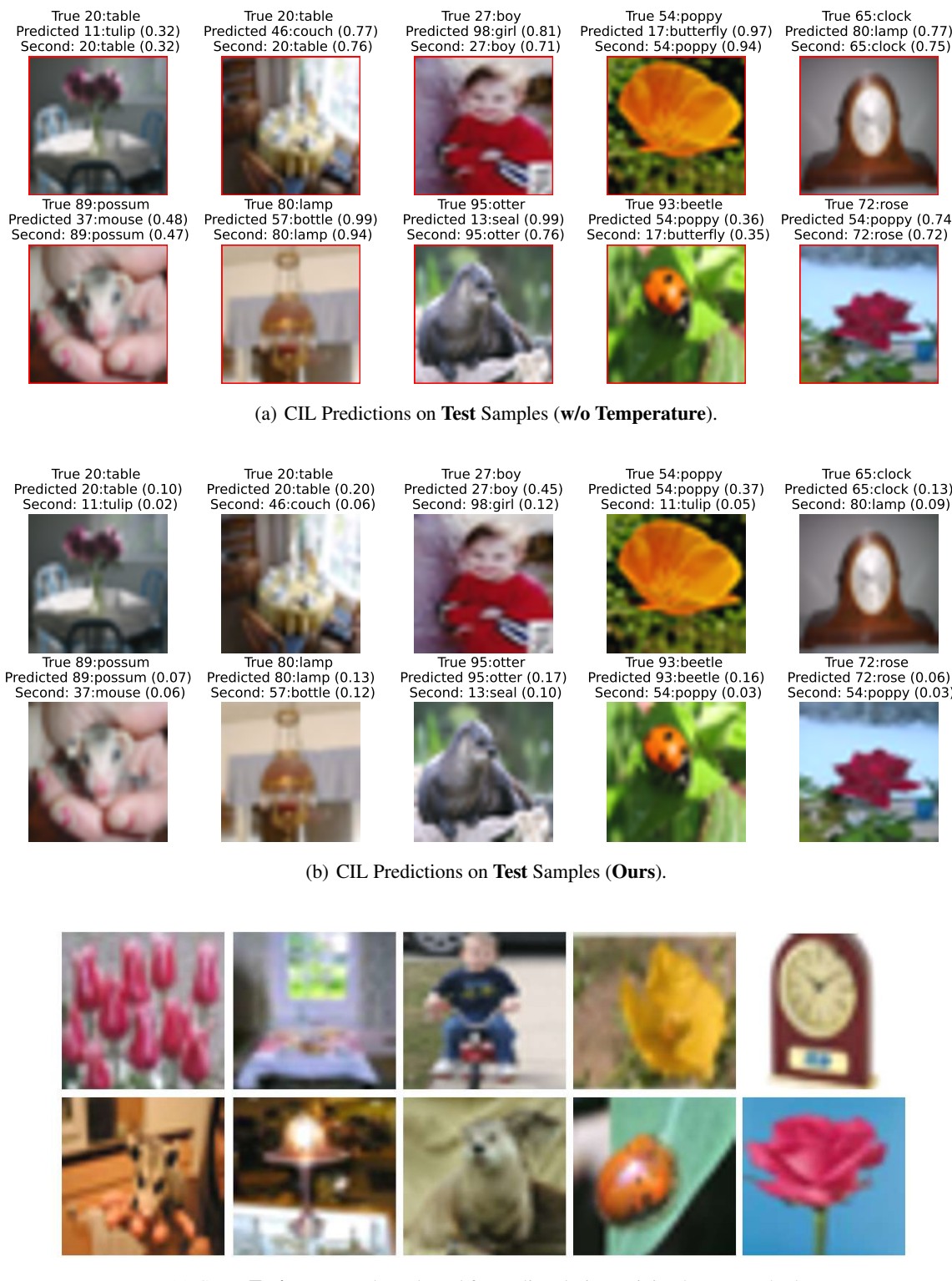

(a) CIL Predictions on **Test** Samples (**w/o Temperature**).

(b) CIL Predictions on **Test** Samples (**Ours**).

(c) Some **Train**-set samples selected for scaling during training by our method.

*Figure 10.* Qualitative examples in the CIL setting. For each sample, we present the true label and predicted label along with the corresponding predicted probability. Additionally, we display the second-highest probability and its associated label. The overall figure demonstrates how independent experts can make incorrect predictions with high confidence, compared to our method's predictions on the same test samples. We also include examples of training samples selected for scaling during training.

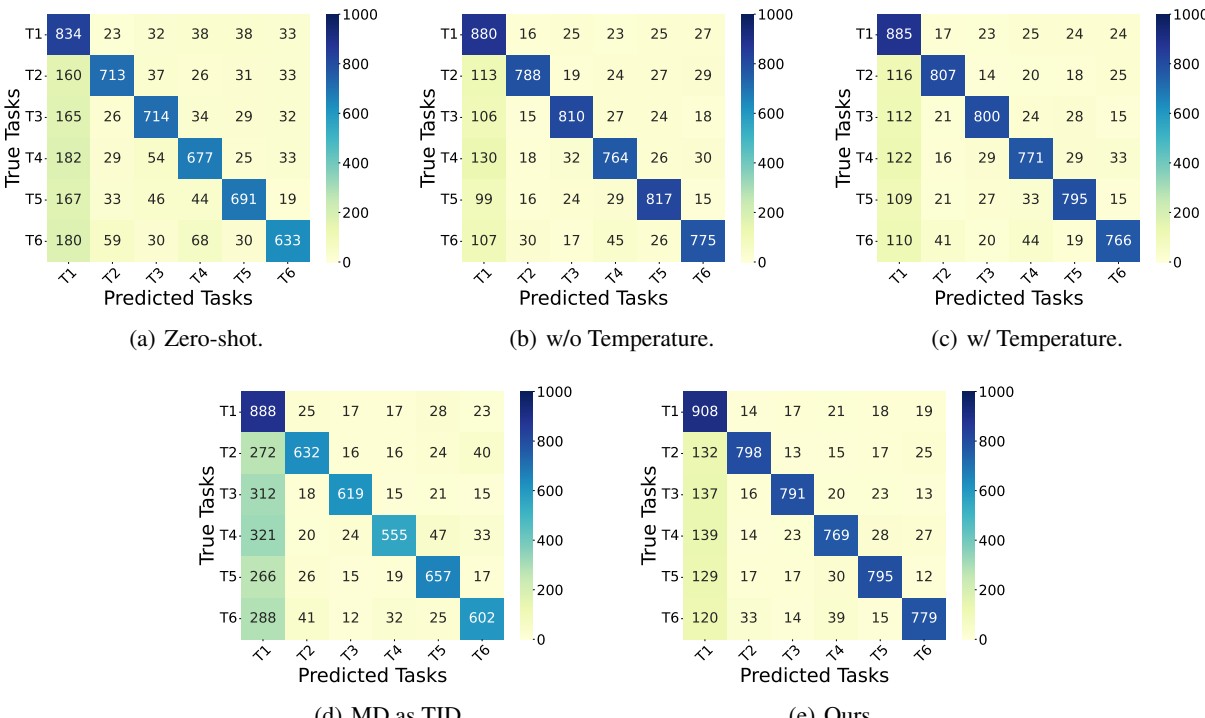

(a) Zero-shot.     (b) w/o Temperature.     (c) w/ Temperature.

(d) MD as TID.     (e) Ours.

*Figure 11.* Comparison of different components of our method on the TinyImageNet dataset with 5 steps. For TinyImageNet, the initial step (i.e., task one) contains a total of 100 classes and 5,000 test samples (task one value scaled to 1000 to match with other tasks). There is a bias towards Task One due to the class imbalance across the other tasks.

*Table 6.* Training parameters.

| Method | Parameters ↓ |
|---|---|
| LwF | 149.6M |
| LwF-VR | 149.6M |
| ZSCL | 149.6M |
| MoE-Adapters | 59.8M |
| Ours | 43.6M |

*Table 7.* Effect of $\tau$ in Equation (4).

| $\tau_{\mathbf{UB}}$ | $\tau_{\mathbf{LB}}$ | **Avg** | **Last** |
|---|---|---|---|
| 1.0 | 1.0 | 85.57 | 77.88 |
| 1.0 | 0.8 | 87.28 | 80.6 |
| 1.0 | 0.7 | 86.44 | 79.7 |
| 1.3 | 0.7 | 84.37 | 76.27 |

*Table 8.* Effect of $\zeta$ in Equation (8).

| **Value** | **Avg** | **Last** |
|---|---|---|
| $\zeta=1.0$ | 85.92 | 78.55 |
| $\zeta=0.1$ | 86.21 | 79.02 |
| $\zeta=0.01$ | 87.28 | 80.6 |
| $\zeta=0.001$ | 85.69 | 77.72 |

*Table 9.* Effect of threshold in Equation (3).

| Value | Avg | Last |
|-------|-------|-------|
| 95 | 87.29 | 80.99 |
| 90 | 87.28 | 80.60 |
| 80 | 86.86 | 80.59 |
| 50 | 85.41 | 78.93 |

### D.3. Sensitivity Analysis of Key Hyperparameters

We present a sensitivity analysis of the key hyperparameters in our method. The experiments are conducted under the same setting as Section 4. In Tables 7 and 8 we report on CIFAR100 with 10 steps.

**Effect of $\tau$ in Equation (4).** In the table 7, when $\tau_{UB} == \tau_{LB}$, the experts apply no adaptive confidence scaling to semantically similar samples, leading to reduced performance. Setting $\tau_{LB} = 0.8$ allows for effective regularization via confidence-scaled gradients from earlier experts, resulting in improved performance. However, further lowering $\tau_{LB}$ (e.g., to 0.7) causes the prior to overly dominate the posterior of the current expert, introducing confusion and reducing effectiveness. We also show that increasing $\tau_{UB}$ leads to lower performance, which is expected, as it reduces the confidence for unfiltered classes and negatively impacts in-distribution classes. Overall, the results demonstrate that the method is robust to reasonable variations in this hyperparameter, with a clear performance peak when moderate scaling is applied.

**Effect of $\zeta$ in Equation (8).** In the table 8, we evaluate the effect of $\zeta$ on softmax weighting. Reducing $\zeta$ improves OOD detection and overall performance compared to no weighting ($\zeta = 1.0$). However, setting it too low (e.g., 0.001) degrades performance, likely due to overly aggressive down-weighting of uncertain samples. These results indicate that our method is robust to this hyperparameter, with $\zeta = 0.01$ providing the best balance.

**Effect of threshold in Equation (3).** Table 9 presents a sensitivity analysis of the threshold value in Equation (3). As discussed in Section 3.2, our threshold mechanism is adaptive per task expert, ensuring that only high-confidence samples for the corresponding expert are filtered. From Table 9, we observe that using a threshold between the 80th and 95th percentiles effectively adapts to key shortcut samples and thus improves learning. However, lowering the threshold too much hampers performance, as an excessive number of samples are filtered out as shortcut samples. We therefore recommend choosing a value between the 80th and 95th percentile for balanced performance across benchmarks.

### D.4. Impact of MD on TIL vs CIL

We describe our distribution-aware weighting mechanism in Section 3.3. This mechanism further supports our experts during prediction. However, as noted in Equation (8), the weighting is sample-specific rather than task-specific, which ensures that within-task predictions are not hampered and enables better out-of-distribution detection. In Table 10, we show that with and without the MD component, our method's performance on task-incremental learning (task identity always available) remains the same. On the other hand, Table 11 demonstrates that in class-incremental learning, where the task ID is unavailable during prediction (making it more challenging than TIL), our MD component helps achieve improved performance, as expected.

*Table 10.* TIL Results on CIFAR-100 – 10 Steps

| Tasks | 1 | 2 | 3 | 4 | 5 | 6 | 7 | 8 | 9 | 10 |
|-------|-------|-------|-------|-------|-------|-------|-------|-------|-------|-------|
| w/ MD | 98.30 | 95.90 | 93.20 | 95.70 | 94.90 | 98.30 | 94.10 | 97.30 | 96.60 | 96.10 |
| w/o MD | 98.30 | 95.90 | 93.20 | 95.70 | 94.90 | 98.30 | 94.10 | 97.30 | 96.60 | 96.10 |

*Table 11.* CIL Results (Last Accuracy) on CIFAR-100 – 10 Steps

| Tasks | 1 | 2 | 3 | 4 | 5 | 6 | 7 | 8 | 9 | 10 |
|-------|-------|-------|-------|-------|-------|-------|-------|-------|-------|-------|
| w/ MD | 82.1 | 82.1 | 63.0 | 78.0 | 82.5 | 89.6 | 76.1 | 85.0 | 84.4 | 83.2 |
| w/o MD | 79.20 | 81.20 | 56.90 | 80.10 | 81.20 | 88.50 | 73.10 | 83.10 | 83.70 | 78.10 |

*Table 12.* Per class Gaussian (CIFAR100 with 10 steps Last Accuracy)

| MD Variant | 1 | 2 | 3 | 4 | 5 | 6 | 7 | 8 | 9 | 10 |
|---|---|---|---|---|---|---|---|---|---|---|
| Single Gaussian (ours) | 82.1 | 82.1 | 63.0 | 78.0 | 82.5 | 89.6 | 76.1 | 85.0 | 84.4 | 83.2 |
| Per-class GMM | 82.0 | 82.5 | 61.2 | 78.9 | 81.7 | 90.9 | 75.1 | 80.6 | 83.3 | 77.7 |
| w/o Gaussian | 79.20 | 81.20 | 56.90 | 80.10 | 81.20 | 88.50 | 73.10 | 83.10 | 83.70 | 78.10 |

*Table 13.* MD vs Router

| Method | GPU ↓ | Times ↓ |
|---|---|---|
| DDAS Router | 2461 MiB | 0.21 seconds |
| MD | 392 MiB | 0.0016 seconds |

## D.5. Single vs Per-Class Gaussian

In Section 3.3 we introduce our multivariate Gaussian, which acts as MD weight per sample for an expert. Our expert communication alone is sufficient to differentiate OOD samples, and the single multivariate Gaussian further supports effective OOD detection. We additionally experimented with using per-class Gaussians (see Table 12). However, as the results show, the simpler per-task Gaussian already provides strong OOD signals, and switching to per-class Gaussians does not significantly boost performance. This behavior is expected. Because our experts are independently optimized for each task (without parameter sharing), the per-task Gaussian naturally aligns with the expert predictions. Consequently, introducing additional per-class Gaussians results in distributions that heavily overlap with the existing expert outputs, thereby providing limited benefit. Moreover, the per-task setup is both more memory-efficient and computationally efficient.

## D.6. MD vs Router

As described in Section 3.3, our distribution-aware weighting (i.e., MD) mechanism leverages learned visual embeddings and is applied during inference as Equation (7). This process is significantly lighter than traditional routing mechanisms. In Table 13, we compare the computational overhead of our MD mechanism against the router used by Yu et al. (2024).

## D.7. Ablation on different model architecture

We conduct an ablation study using a different CLIP model variant, RN50x16, as the image encoder. We keep all other settings the same and apply LoRA adapters to the ResNet at the pooled feature level, after the standard convolutional layers and attention pooling. We compare our method with the MoE-based baseline for a fair evaluation. As shown in Table 14, even with this substantially different image-encoder architecture, our method exhibits performance trends similar to those in Table 1. This demonstrates the robustness of our method across different architectures.

*Table 14.* Ablation with CLIP using a ResNet-50 image encoder.

| Method | CIFAR100 (10 Steps) |
|---|---|
| MoE-Adapters | 33.57 |
| Ours | 42.98 |

## E. Impact of Multimodality on learning

**Setting.** We use the Colorized-MNIST (Jayaneetha, 2020) dataset, which contains handwritten digits (0–9) overlaid on colorized backgrounds. There are three background colors available. To study the impact of background cues when textual information is not available, we divide the **training set** into three groups: digit 0 is paired exclusively with a red background, digit 1 with blue, and digit 2 with green. Since only three background colors are available, we restrict the study to these three digits. In the **test set**, however, each digit appears against all three background colors. We use a pre-trained ResNet-50 and a pre-trained CLIP model for this experiment. We use the prompt template for CLIP: ``a bad photo of a handwritten digit {}''.

**Analysis.** We evaluate model performance using classification accuracy, averaged over three runs. CLIP achieves an average accuracy of 81.67%, while ResNet-50 achieves only 31.05%. Figure 12 shows the qualitative results of this experiment.

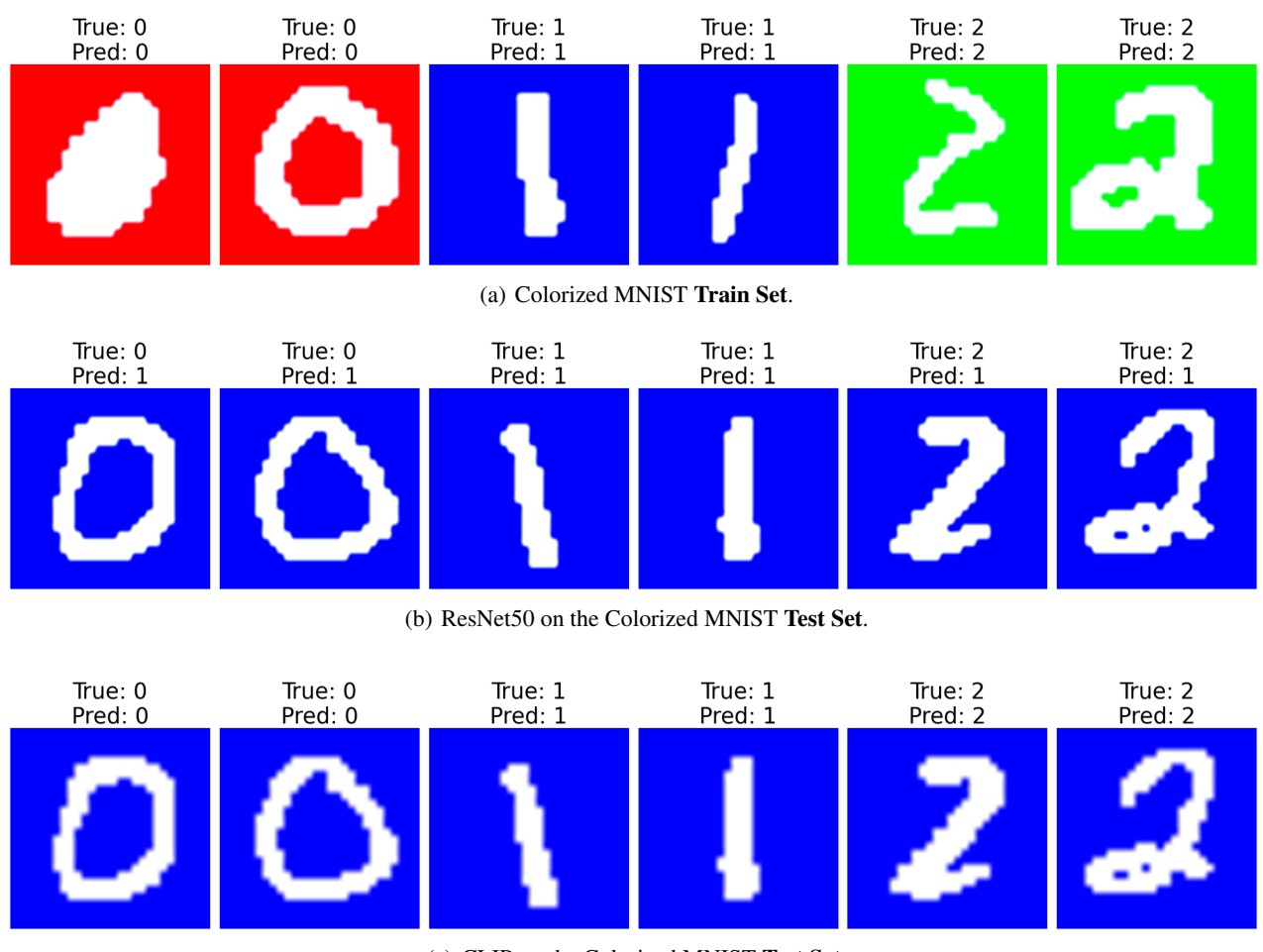

(a) Colorized MNIST **Train Set**.

(b) ResNet50 on the Colorized MNIST **Test Set**.

(c) CLIP on the Colorized MNIST **Test Set**.

*Figure 12.* Qualitative examples on the Colorized MNIST dataset demonstrate that CLIP, by aligning information from language modalities, helps the model learn more robust features.

Both models perform well on the training set (Figures 12(a), achieving on average 99.9% accuracy. However, on the test set, where digits appear with mix-and-match background colors (i.e., digit 0 with red, blue, or green, which differs from the training set), ResNet-50 relies on background cues and frequently misclassifies digits (Figure 12(b)). In contrast, CLIP leverages its text prompts to correctly identify the digit shape, demonstrating robustness to background variations (Figure 12(c)).

Overall, this experiment highlights the advantage of multimodal learning. Aligning visual and linguistic information helps models learn more semantically meaningful and robust features, enabling better generalization beyond superficial cues.

## F. Sub-population Shift Incremental Learning

**Setting.** Following Section 4.2 in the main paper, we provide additional details on the Sub-population Shift. For the sub-population shift, we use the BREEDS benchmark (Santurkar et al., 2021) following the protocol outlined in Liang et al. (2022). For Entity-13 with 10 steps, each step receives one unseen subclass. For 13 steps, we randomly sample 10 out of 13 classes, with each receiving one unseen subclass sample. For Entity-30 with 8 steps, 15 randomly sampled classes from the 30 superclasses receive one unseen subclass at each step. Similarly, for Entity-30 with 15 steps, 8 randomly sampled classes from the 30 superclasses receive one unseen subclass at each step. This protocol mimics two real-life situations: 1) when each superclass receives at least one sample, we refer to it as even; and 2) when part of the superclasses receives samples, we refer to it as an uneven update. We utilize the maximum confidence score for the final prediction, and the number of experts

*Table 15.* Performance comparison in the sub-population shift incremental learning setting with other baselines on Entity-13 with 10 steps (Even Update). We report both "seen", "unseen" (inside parentheses), and "All" accuracy in % over the last eight incremental steps. $\Delta$ denotes performance improvement with respect to the second-best method.

| | | | | Entity 13 Benchmark 10 Steps (Even Update) | | | | | |
|---|---|---|---|---|---|---|---|---|---|
| **Model** | 3 | 4 | 5 | 6 | 7 | 8 | 9 | 10 | **All** |
| CLIP Zero-shot | 67.04 | 69.11 | 69.46 | 70.15 | 72.19 | 71.84 | 72.09 | 72.02 | 71.04 |
| ZSCL (Zheng et al., 2023) | 86.20 (75.60) | 80.70 (71.74) | 79.69 (71.16) | 84.84 (71.35) | 87.02 (75.25) | 82.35 (84.0) | 83.96 (75.38) | 80.66 | 77.78 |
| MoE-Adapters (Yu et al., 2024) | 89.11 (71.86) | 83.53 (68.87) | 82.34 (67.87) | 85.04 (70.96) | 85.24 (71.38) | 81.92 (80.46) | 74.91 (67.69) | 71.75 | 77.02 |
| RAIL (Xu et al., 2024) | 86.86 (70.98) | 86.62 (71.29) | 85.88 (71.66) | 86.16 (72.04) | 85.86 (71.99) | 85.82 (72.06) | 85.85 (72.02) | 84.56 | 79.30 |
| **Ours** | 93.08 (80.90) | 91.28 (81.17) | 89.13 (81.75) | 88.47 (80.34) | 88.77 (77.99) | 87.12 (83.77) | 87.57 (79.23) | 87.13 | 84.96 $\Delta$+7.14% |

*Table 16.* Performance comparison in the sub-population shift incremental learning setting with other baselines on Entity-30 with 8 steps (Uneven Update). We report both "seen", "unseen" (inside parentheses), and "All" accuracy in % over the last eight incremental steps. $\Delta$ denotes performance improvement with respect to the second-best method.

| | | | | Entity 30 Benchmark 8 Steps (Uneven Update) | | | | | |
|---|---|---|---|---|---|---|---|---|---|
| **Model** | 1 | 2 | 3 | 4 | 5 | 6 | 7 | 8 | **All** |
| CLIP Zero-shot | 67.13 | 64.22 | 67.09 | 65.06 | 66.66 | 65.92 | 68.16 | 65.76 | 66.50 |
| ZSCL (Zheng et al., 2023) | 78.30 (51.08) | 86.94 (76.58) | 86.96 (68.90) | 87.29 (75.69) | 87.47 (64.97) | 85.69 (72.93) | 86.49 (56.40) | 82.82 | 75.65 |
| MoE-Adapters (Yu et al., 2024) | 87.47 (51.67) | 88.08 (67.66) | 79.23 (51.81) | 80.93 (63.36) | 80.99 (57.06) | 75.55 (56.93) | 78.69 (39.87) | 67.15 | 70.23 |
| RAIL (Xu et al., 2024) | 90.10 (66.12) | 87.52 (66.44) | 86.98 (66.31) | 85.81 (66.62) | 85.31 (66.61) | 85.40 (66.61) | 85.07 (65.76) | 84.59 | 77.23 |
| **Ours** | 94.03 (66.81) | 91.63 (78.33) | 87.29 (71.94) | 86.69 (78.36) | 84.61 (69.55) | 85.52 (72.45) | 84.34 (50.93) | 83.64 | 79.85 $\Delta$+3.4% |

is set to 11. We match the expert count with Yu et al. (2024) for a fair comparison. Other details follow those mentioned in Zheng et al. (2023); Yu et al. (2024).

**Analysis.** Table 15 presents the results for Entity-13 with 10 steps. This represents an even update situation, and we observe that our method consistently outperforms other baselines by a significant margin. Table 16 demonstrates an uneven update situation. Finally, Tables 17 and 18 illustrate even more complex scenarios, where the step size is larger and the updates are uneven. Even in these more challenging situations, our method demonstrates its superiority compared to other baselines. These results further validate the robustness of our method.

# G. Cross-domain Task-Agnostic Incremental Learning

**Setting.** Following Xu et al. (2024), we also compared our method on the Cross-domain Task-Agnostic Incremental Learning (X-TAIL) benchmark, as discussed in the main paper, Section 4, and Figure 5. The X-TAIL benchmark comprises a total of 11 tasks, with each task corresponding to an individual dataset rather than a single dataset divided into multiple tasks. This ensures that each task originates from a distinct domain. This benchmark represents a cross-domain variation of the Multi-domain Task Incremental Learning (MTIL) (Yu et al., 2024) benchmark. The 11 datasets used are Aircraft (Maji et al., 2013), Caltech101 (Fei-Fei et al., 2004), CIFAR100 (Krizhevsky, 2009), DTD (Cimpoi et al., 2014), EuroSAT (Helber et al., 2019), Flowers (Nilsback & Zisserman, 2008), Food (Bossard et al., 2014), MNIST (Deng, 2012), OxfordPet (Parkhi et al., 2012), StanfordCars (Krause et al., 2013), and SUN397 (Xiao et al., 2010). Specifically, the Aircraft dataset contains 100 classes, Caltech101 has 101 classes, CIFAR100 includes 100 classes, DTD has 47 classes, EuroSAT 10 classes, Flowers 102 classes, Food 101 classes, MNIST 10 classes, OxfordPet 37 classes, StanfordCars 196 classes, and SUN397 comprises 397 classes. Together, these 11 tasks encompass a total of 1,201 classes. For this benchmark, we used the same settings mentioned in the main paper. We utilize the metrics proposed by Xu et al. (2024) to evaluate our method on the X-TAIL benchmark. We show results on "**Transfer**," "**Average**," and "**Last**." The "Transfer" metric evaluates the model's zero-shot

*Table 17.* Performance comparison in the sub-population shift incremental learning setting with other baselines on Entity-13 with 13 steps (Uneven Update). We report both "seen", "unseen" (inside parentheses), and "All" accuracy in % over the odd incremental steps. $\Delta$ denotes performance improvement with respect to the second-best method.

| | | | | Entity 13 Benchmark 13 Steps (Uneven Update) | | | | | |
|---|---|---|---|---|---|---|---|---|---|
| **Model** | 1 | 3 | 5 | 7 | 9 | 11 | 13 | **All** |
| CLIP Zero-shot | 72.34 | 71.67 | 67.91 | 70.33 | 71.84 | 72.62 | 71.96 | 71.25 |
| ZSCL (Zheng et al., 2023) | 84.57 (65.48) | 85.50 (68.48) | 84.42 (72.7) | 81.64 (70.26) | 83.85 (74.5) | 79.36 (79.4) | 81.70 | 76.26 |
| MoE-Adapters (Yu et al., 2024) | 91.17 (62.76) | 82.55 (65.22) | 84.71 (58.47) | 80.08 (67.5) | 83.25 (74.5) | 79.34 (74.69) | 79.35 | 74.55 |
| RAIL (Xu et al., 2024) | 91.40 (70.84) | 86.41 (70.57) | 86.33 (71.34) | 85.54 (71.87) | 85.68 (72.37) | 85.47 (72.26) | 84.50 | 79.07 |
| **Ours** | 95.99 (75.84) | 92.85 (79.1) | 89.07 (81.89) | 87.56 (81.3) | 87.52 (79.7) | 86.10 (80.69) | 86.48 | 83.81 $\Delta$+5.9% |

*Table 18.* Performance comparison in the sub-population shift incremental learning setting with other baselines on Entity-30 with 15 steps (Uneven Update). We report both "seen", "unseen" (inside parentheses), and "All" accuracy in % over the odd incremental steps. $\Delta$ denotes performance improvement with respect to the second-best method.

| | | | | | | Entity 30 Benchmark 15 Steps (Uneven Update) | | | | |
|---|---|---|---|---|---|---|---|---|---|---|
| Model | 1 | 3 | 5 | 7 | 9 | 11 | 13 | 15 | All |
| CLIP Zero-shot | 74.76 | 64.07 | 67.50 | 64.28 | 66.75 | 65.50 | 67.91 | 65.61 | 66.76 |
| ZSCL (Zheng et al., 2023) | 79.08 (53.51) | 86.75 (62.27) | 89.06 (66.77) | 84.70 (71.43) | 85.11 (66.79) | 84.40 (66.06) | 85.79 (49.12) | 81.76 | 74.24 |
| MoE-Adapters (Yu et al., 2024) | 95.84 (65.92) | 65.94 (42.39) | 79.21 (52.17) | 68.44 (51.03) | 65.62 (40.20) | 62.85 (43.62) | 56.65 (14.5) | 53.90 | 59.99 |
| RAIL (Xu et al., 2024) | 90.54 (66.06) | 87.12 (66.12) | 85.65 (66.07) | 84.90 (66.25) | 84.82 (66.26) | 85.00 (66.88) | 84.71 (66.32) | 84.25 | 76.27 |
| **Ours** | 96.56 (70.12) | 91.25 (75.58) | 87.77 (73.75) | 85.87 (78.34) | 85.58 (72.83) | 83.99 (73.06) | 84.17 (53.25) | 81.82 | 79.58 $\Delta$+4.34% |

*Table 19.* Comparison on X-TAIL benchmark with other SOTA baselines. Overall, our method showcases superior adaptability to cross-domain tasks without the help of **reference dataset** or **replay buffer**. **Bold** denotes best and underline denotes second-best.

| | Method | Aircraft | Caltech101 | CIFAR100 | DTD | EuroSAT | Flowers | Food | MNIST | OxfordPet | Cars | SUN397 | Average |
|---|---|---|---|---|---|---|---|---|---|---|---|---|---|
| **Transfer** | ZSCL (Zheng et al., 2023) | | 63.7 | 37.2 | 32.1 | 15.8 | 60.1 | 82.9 | 32.1 | 82.6 | 53.3 | 53.6 | 51.3 |
| | MoE-Adapters (Yu et al., 2024) | | 64.2 | 35.9 | 32.9 | 17.3 | 60.6 | 86.6 | 23.0 | 87.2 | 63.7 | 57.1 | 52.9 |
| | RAIL (Xu et al., 2024) | | **69.7** | 37.3 | 36.5 | 36.6 | 60.7 | 84.0 | **46.6** | 86.7 | **66.1** | **62.5** | 58.7 |
| | Ours | | 63.25 | **39.29** | **36.73** | **51.52** | **71.19** | **88.58** | 43.00 | **88.9** | 63.77 | 60.79 | **60.70** |
| **Average** | ZSCL (Zheng et al., 2023) | 33.4 | 57.9 | 41.0 | 37.7 | 20.3 | 68.1 | 84.0 | 36.1 | 82.0 | 57.7 | 55.2 | 52.1 |
| | MoE-Adapters (Yu et al., 2024) | 42.4 | 66.4 | 55.3 | 49.0 | 38.3 | 74.9 | 86.2 | 46.7 | 87.4 | 66.2 | 58.4 | 61.0 |
| | RAIL (Xu et al., 2024) | 45.0 | **88.8** | 57.8 | 56.8 | 66.2 | 81.0 | 85.2 | 63.4 | 87.8 | **68.9** | **64.7** | 69.6 |
| | Ours | **53.15** | 78.73 | **68.45** | **58.42** | **80.45** | **83.32** | **90.01** | 63.01 | **90.18** | 67.83 | 62.50 | **72.37** |
| **Last** | ZSCL (Zheng et al., 2023) | 31.4 | 59.6 | 43.9 | 39.7 | 28.4 | 71.6 | 86.4 | 40.7 | 82.6 | 77.0 | 70.8 | 57.5 |
| | MoE-Adapters (Yu et al., 2024) | 41.8 | 66.2 | 59.5 | 53.7 | 45.9 | 84.3 | 85.8 | 86.8 | 87.7 | 76.2 | 71.7 | 69.1 |
| | RAIL (Xu et al., 2024) | 45.2 | **94.4** | 74.7 | 70.7 | 87.3 | **97.9** | 86.5 | 92.8 | 91.9 | 81.7 | 76.7 | 81.8 |
| | Ours | **52.87** | 89.46 | **85.34** | **73.94** | **97.3** | 93.43 | **92.0** | **98.05** | **93.76** | **86.12** | **79.66** | **85.63** |

transfer capability on unseen data, while "Last" measures the model's ability to retain and utilize historical knowledge. "Average" serves as a composite metric, representing the mean performance on "Transfer" and "Last". We use publicly available results by Xu et al. (2024) from the OpenReview discussion, which additionally includes results on CIFAR-100 in X-TAIL, reflecting a more complete and standardized evaluation setting.

**Analysis.** Table 19 presents a performance comparison with state-of-the-art methods across Transfer, Average, and Last accuracy metrics, where higher values indicate better performance. Our method consistently achieves the highest scores in all three metrics across multiple tasks, highlighting its robustness. Notably, unlike existing baselines that rely on a `reference dataset`, `replay buffer`, or CLIP's zero-shot capabilities to mitigate out-of-distribution challenges, our approach operates entirely without such dependencies. This independence enhances its practicality and compatibility with real-world deployment. Overall, the superior cross-domain adaptability of our method underscores its strong generalization, stability, and robustness.

# H. Limitations

While our method benefits from a dual perspective, such as adaptive temperature scaling and multivariate Gaussian-based MD for two different kinds of uncertainty, epistemic uncertainty (i.e., model uncertainty) and aleatoric uncertainty (i.e., data uncertainty), it still has some limitations. One of the main limitations is that our method only accounts for past experiences while optimizing the current expert and allocates task-specific capacity progressively. We deliberately avoid parameter sharing or merging during training to preserve strict isolation and eliminate forgetting. The framework is nonetheless compatible with post-hoc consolidation strategies (e.g., merging based on distributional similarity, such as Mahalanobis overlap) or sparsified expert selection, which we leave for future work.

# I. Implementation Details

Throughout our experiments, we use the CLIP (Radford et al., 2021) with a ViT-B/16 (Dosovitskiy et al., 2021) image encoder. For expert implementation, we utilize a LoRA (Hu et al., 2022) adapter, which is the only trainable component of

our method, while the backbone remains frozen. We use the adapter for both text and vision components. The number of experts is set equal to the task size for all benchmarks. We apply label smoothing (Müller et al., 2019) and the AdamW optimizer (Loshchilov & Hutter, 2019), training our method for 1,000 iterations. We use the 90th percentile as the threshold for the top-k confidence set with $\tau_{UB}$ and $\tau_{LB}$ ranging from 0.8 to 1.0, and $\zeta$ is set to 0.01. We follow Zheng et al. (2023), Yu et al. (2024), Novack et al. (2023), and Xu et al. (2024) for other settings.

### I.1. Workstation Details.

Our workstation is equipped with an AMD Ryzen Threadripper PRO 5995WX processor (64 physical cores), NVIDIA RTX A6000 GPUs (48 GB each), and 256 GB of system RAM.

## J. Source Code

The implementation code is publicly available at: `https://github.com/alforhad/MoE-Confidence`

