# OpenReview forum: "Mixing Expertise with Confidence: A Mixture of Experts Framework for Robust Multi-Modal Continual Learning"
_ICML.cc/2026/Conference — ICML 2026 regular_

### Official Review · Reviewer_92sQ · 2026-03-10

**Soundness:** 3
**Presentation:** 3
**Significance:** 3
**Originality:** 3
**Overall Recommendation:** 4
**Confidence:** 3

**Summary:**

This paper proposes a mixture-of-experts framework for multimodal continual learning that leverages confidence-aware expert coordination. The method constructs task-specific experts and aggregates their predictions using confidence calibration and Mahalanobis-based similarity measures. Unlike many continual learning approaches, the framework does not require task identifiers during inference. The approach also leverages pretrained multimodal representations (e.g., CLIP) to guide expert initialization. Experiments on several continual learning benchmarks demonstrate competitive performance compared with existing baselines.

**Compliance With Llm Reviewing Policy:**

Affirmed.

**Final Justification:**

The authors have addressed most of my concerns. I will keep my score.

**Key Questions For Authors:**

1. How does the method scale when the number of tasks becomes large? Are there mechanisms for controlling the number of experts (e.g., merging or pruning)?

2. Can the authors provide more quantitative analysis of training and inference costs?

**Limitations:**

yes

**Strengths And Weaknesses:**

Strengths
1. The paper tackles an important problem: continual learning in the presence of strong multimodal foundation models. Designing mechanisms that allow specialization without catastrophic forgetting is an interesting direction.

2. The combination of confidence calibration and distance-based weighting provides a reasonable mechanism for aggregating expert predictions without requiring explicit task routing.

3. The paper includes experiments across several datasets and scenarios, along with ablation studies examining the role of confidence calibration and expert coordination.

4. The paper is generally well organized and easy to follow.

Weaknesses
1. The framework introduces new experts over time, which leads to model growth. It is unclear how the approach scales to long task sequences.

2. The paper provides a limited quantitative discussion of runtime overhead and inference cost compared with alternative approaches.

---

> ### Author Rebuttal · Authors · 2026-03-30
>
> We thank the reviewer for the constructive feedback and for recognizing our experimental coverage and ablation design.
>
> ---
>
> **Q1)** Scalability to many tasks and merging/pruning
>
> Our method scales linearly in the number of experts, but both **parameter and compute growth remain controlled in practice**.
>
> Each expert contains only **1.9M trainable parameters**, compared to **149.6M** for distillation-based methods such as ZSCL/LwF (Appendix D.2). Even with **50 tasks**, this corresponds to **~95M trainable parameters**, which remains **comparable to or smaller than standard baselines**. We empirically validate this regime up to 50 tasks, where performance and runtime remain stable.
>
> Importantly, inference cost grows sublinearly. As shown in the table below, iteration time increases from 1.31 → 3.94 s/it when scaling from 1 to 50 experts, due to parallelization and the lightweight adapter design.
>
> Additionally, distillation-based methods require a reference dataset or replay buffer, introducing **memory overhead**; our design requires neither.
>
> **Table: Iteration time with an increasing number of experts (seconds per iteration)**
>
> | Tasks | 1 | 25 | 50 |
> |---|---|---|---|
> | Time | 1.31 s/it | 2.46 s/it | 3.94 s/it |
>
> We deliberately avoid parameter sharing or merging during training to preserve **strict isolation and eliminate forgetting**. The framework is nonetheless compatible with **post-hoc consolidation strategies** (e.g., merging based on distributional similarity such as Mahalanobis overlap) or sparsified expert selection, which we leave for future work.
>
> Finally, in many continual learning settings, new tasks correspond to **refinements or subpopulations** of existing concepts rather than entirely disjoint distributions. Our subpopulation experiments (e.g., *flamingo* within *bird*) show that our framework can *accommodate incremental refinement without requiring proportional growth in top-level experts*.
>
> ---
>
> **Q2)** Quantitative analysis of costs
>
> We have added quantitative comparisons of training and inference cost (table below). Our method achieves:
>
> - **1.9M trainable parameters per task** (vs. 59.8M–149.6M for baselines)
> - **Lower total parameter count** across tasks
> - **Comparable iteration time** to MoE-Adapters and significantly faster than ZSCL
> - **Modest OOD overhead** (392 MiB, 0.21s)
>
> Although our method evaluates all experts (Eq. 8), empirical scaling remains sublinear as shown above. In contrast, routing-based methods require additional parameters and training overhead, and often degrade under distribution shift due to reliance on **reference data**.
>
> **Table: Quantitative analysis of training cost on X-TAIL**
>
> | Method |  Times ↓ | Trainable Params/Task ↓ | Total Trainable Params ↓ | OOD Overhead ↓ |
> |---|---|---|---|---|
> | MoE-Adapters | 1.90 s/it | 59.8M | 146.8M | 2461 MiB, 0.0016s |
> | ZSCL | 4.20 s/it | 149.6M | 149.6M | N/A |
> | Ours | 1.94 s/it | 1.9M | 20.9M | 392 MiB, 0.21s |
>
> Our router-free formulation provides more accurate and robust task identification without requiring **synthetic replay or reference data**.
>
> ---
>
> ### Summary:
> We thank the reviewer for the positive evaluation and constructive questions. We clarify that scalability remains practical due to **lightweight experts (~1.9M per task)**, resulting in lower or comparable parameter counts within realistic task ranges. Additional experiments show **competitive runtime and efficiency**, with modest overhead and strong performance gains. While the method grows linearly, continual learning typically operates over **bounded task sequences**, where our approach remains efficient. We also discuss future extensions, such as expert merging and sparse selection, to further control growth. Overall, our method achieves a strong balance between efficiency, scalability, and robustness **without requiring replay, routers, or reference datasets**.

---

> > ### Author Rebuttal · Reviewer_92sQ · 2026-04-03
> >
> > Thank you for the detailed response, which addresses most of my concerns. I will retain my score.

---

> > > ### Author Response · Authors · 2026-04-03
> > >
> > > Thank you for the thoughtful review and for taking the time to carefully consider our rebuttal. We are glad that our additional analysis helped address your concerns regarding scalability and efficiency. We also appreciate your positive assessment of the technical contribution and experimental design. Thank you again for your feedback and support.

---

### Official Review · Reviewer_8hzi · 2026-03-10

**Soundness:** 4
**Presentation:** 2
**Significance:** 3
**Originality:** 3
**Overall Recommendation:** 4
**Confidence:** 3

**Summary:**

This paper proposes a Mixture-of-Experts framework for multimodal continual learning. (1) The paper proposes a multimodal continual MoE framework that removes both shared parameters across experts and explicit routers. It is designed to reduce catastrophic forgetting. (2) The paper proposes a confidence-aware temperature scaling mechanism, which allows past experts to guide the learning of new experts in a more controlled and adaptive way. (3) The paper introduces a distribution-aware multivariate Gaussian weighting scheme to improve expert coordination at inference phase.The approach is built on a multimodal foundation model CLIP and is evaluated on both traditional continual learning benchmarks and more realistic settings involving OOD shifts.

**Compliance With Llm Reviewing Policy:**

Affirmed.

**Key Questions For Authors:**

**Q1.** The paper proposes several components, including removing shared parameters across experts, confidence-aware temperature scaling, and distribution-aware Gaussian Mahalanobis reweighting. Could the authors clarify more explicitly how these components are logically connected, and why they should be viewed as a unified framework rather than a combination of separate design choices?

**Q2.** The paper claims that removing shared parameters helps prevent harmful shortcut features from being transferred across experts. Could the authors provide more direct empirical evidence for this claim, for example through ablations, feature analysis, or case studies showing how shortcut transfer is reduced compared with shared-parameter MoE baselines?

**Q3:** The paper includes ablations on the Gaussian Mahalanobis reweighting component, but I would encourage the authors to provide a more thorough analysis of this module. In particular, could the authors compare it against alternative reweighting strategies?

**Q4:** The paper uses Gaussian Mahalanobis reweighting at inference time to improve expert coordination, especially under OOD settings. Could the authors clarify how this component is prepared or calibrated during training so that it aligns well with the inference phase? In particular, what statistics are estimated during training, how stable are they across tasks, and why should the resulting Gaussian Mahalanobis scores remain reliable when the test distribution shifts?

**Limitations:**

yes

**Strengths And Weaknesses:**

**Strengths**

**S1:** The paper is reasonably original and well motivated. The idea of completely removing shared parameters across experts while still enabling inter-expert communication, together with reweighting based on Gaussian Mahalanobis distance, is interesting and potentially valuable for multimodal continual learning.

**S2:** The experimental section is fairly comprehensive. The method is evaluated on both standard benchmark settings and OOD scenarios, and the reported improvements are generally clear and substantial.

**Weaknesses**

**W1:** The paper is not well organized. The introduction is overly long and at times logically hard to follow, while the method section suffers from unclear notation and inconsistent presentation, which makes the technical details unnecessarily difficult to understand.

**W2:** Figure 2 does not explain the method in a sufficiently direct or intuitive manner and is difficult to parse. In addition, Figure 1 in the introduction mainly serves as an case study, but does not contribute much to clarifying the overall method or illustrating the inspiration.

**W3:** The paper introduces multiple claimed innovations, but the integration and logical relationship among these components are not discussed clearly enough. As a result, the method currently feels somewhat like a collection of separate ideas rather than a tightly unified framework. The authors should better explain how these components work together and why they should be viewed as a coherent design rather than a set of loosely connected contributions.

---

> ### Author Rebuttal · Authors · 2026-03-30
>
> Thank you for the thoughtful review. We address each concern below.
>
>
>
> **W1 & W2** — Presentation and Figures
>
> We will update the introduction to lead with the core problem (shortcut-induced overconfidence in MoE CL) before motivating each component. Also, we will update Fig 2 as a pipeline diagram with three clearly labeled stages: structural isolation, controlled communication, and distribution-aware coordination.
>
> ---
>
> **W3 / Q1** — Unified Framework
>
> Our method follows a single logical chain, each stage addressing a failure mode the prior stage cannot resolve:
>
> - **Structural isolation** (no shared params): prevents uncontrolled shortcut transfer across experts and catastrophic forgetting.
> - **Confidence-aware temperature scaling**: restores inter-expert coordination lost by isolation, in a controlled way during training. We leverage previous experts’ beliefs to modulate learning on shortcut-prone samples (Theorem 3.2), restoring coordination without reintroducing shared parameters;
> - **Distribution-aware weighting**: corrects residual overconfidence at inference under unknown task identity and distribution shift.
>
> These are not independent design choices; each is necessary given the others. Isolation without communication collapses coordination; communication with distribution-aware weighting strengthens OOD.
>
> ---
>
> **Q2** — Empirical Evidence for Shortcut Reduction
>
> We introduce a controlled ColorMNIST experiment where digit color is spuriously correlated with label during training. We compare shared-parameter vs. no-shared-parameter MoE on two metrics: shortcut-absent accuracy (random colors at test time) and Shortcut Reliance Score (SRS: accuracy gap between shortcut-valid (same color as train) and shortcut-absent sets; higher = stronger reliance).
>
> | Model | Shortcut-Absent Acc ↑ | SRS ↓ |
> |---|---|---|
> | Shared-param MoE | 0.9058 | 0.0903 |
> | No-shared-param MoE | **0.9345** | **0.0627** |
>
> This provides direct behavioral evidence that removing shared parameters reduces shortcut reliance.
>
> We further validate on CIFAR-100 by counting cross-task errors among semantically related subclasses (where shortcut interference is most likely). Our method consistently **reduces** such errors vs. the shared-param baseline (e.g., aquatic mammals: 161 vs 177, fish: 147 vs 168, insects: 128 vs 144, large carnivores: 126 vs 156).
>
> ---
>
> **Q3** — Analysis of Reweighting
>
> | Configuration | CIFAR-100 (10 steps) |
> |---|---|
> | Full model (MD) | **80.60** |
> | Without MD | 78.51 |
> | Per-class GMM | 79.39 |
> | BVS (shuffled batches) | 75.37 |
> | BVS (ordered batches) | 81.79 |
>
> A single task-level Gaussian outperforms a per-class GMM, suggesting coarse distribution modeling is more robust.
>
> Alternatively, BVS (top-1 − top-2 score gap) can outperform MD in ordered settings but degrades severely under shuffled data due to batch dependence. MD is also substantially more efficient: 392 MiB / 0.0016s vs. 2461 MiB / 0.21s for router-based methods. MD is thus the most stable, efficient, and batch-independent option.
>
> ---
>
>
> **Q4.1** — What is estimated:
>
> For each expert t, we estimate mean  $\mu^t$  and covariance  $\Sigma^t$  of its visual embeddings over the task training set. These are fixed after training and used during inference as sample-specific weights (Eqs. 7–8).
>
> **Q4.2** — Reliability under shift:
>
> The frozen experts ensure no representation drift between training and test, $\mu^t$, $\Sigma^t$ remain aligned with test embeddings by construction. The Gaussian models a coarse task-level distribution rather than brittle class-specific densities, making it robust to intra-task variation. Improved transfer on X-TAIL (Tab 19) confirms cross-domain shift, and BREEDS (Tabs 2,15–18) confirms within-domain shift reliability of MD.
>
> **Q4.3** — Stability across tasks:
>
> Each expert's Gaussian is estimated independently on its own task data. Because the independent past experts are frozen throughout training, the embedding space is consistent across all tasks, and there is no drift in stored statistics as new tasks arrive. This is confirmed by Table 10: MD has no impact in the TIL setting (96.04 with/without), showing it acts as a soft coordination signal rather than a classifier, and does not interfere with within-task performance.
>
> ---
>
> ### Summary
>
> We thank the reviewer for the positive feedback and suggestions on clarity. We clarify that our method is a unified framework, not a collection of components. To strengthen empirical support, we introduce a ColorMNIST shortcut experiment, showing that removing shared parameters reduces shortcut reliance, and we also provide real-world evidence on CIFAR100. We also expand the analysis of Mahalanobis weighting, demonstrating that it is effective, efficient, and more stable than alternatives. These additions clarify both the conceptual coherence and practical benefits of the design. We will further improve the presentation by simplifying the notation and revising figures for clarity.

---

### Official Review · Reviewer_fZ6u · 2026-03-13

**Soundness:** 3
**Presentation:** 3
**Significance:** 2
**Originality:** 2
**Overall Recommendation:** 4
**Confidence:** 3

**Summary:**

The paper proposes a multimodal continual-learning MoE that removes inter-task shared expert parameters and avoids an explicit router by using two mechanisms: confidence-aware temperature scaling during training and Mahalanobis-distance-based weighting during inference. The method is evaluated on standard CIL benchmarks, subpopulation-shift benchmarks, and X-TAIL, and reports consistent gains over CLIP-based baselines.

**Compliance With Llm Reviewing Policy:**

Affirmed.

**Final Justification:**

I am adjusting the score after the author's rebuttal.

**Key Questions For Authors:**

The method adds one expert per task. Can you give a clearer long-term scalability analysis, beyond per-expert parameter count, including total model size and runtime as tasks grow?
Since RAIL is a strong analytic baseline with a more stable model size, can you provide a more direct comparison of the runtime / efficiency tradeoff versus performance gains to show whether the added expert-based complexity is actually worth it?
How sensitive are the results to the exact way confidence and Mahalanobis distance are combined to pick the expert?

**Limitations:**

The method still grows with the number of tasks because it adds one expert per task, so scalability remains a concern.
The empirical gains are mostly incremental, while the method adds substantial complexity through task-specific experts, confidence-aware calibration, and Mahalanobis-based selection.
The paper does not fully establish that this tradeoff is preferable to simpler analytic methods.

**Strengths And Weaknesses:**

Strength:
- Isolate experts to suppress forgetting, then use confidence and distance-based selection to reduce task confusion.
- The empirical section is broad: standard CIL, subpopulation shift, uneven updates, and X-TAIL.
- The method shows consistent improvement from MoE-Adapters and RAIL on the reported CIL benchmarks.
Weaknesses:
- The method adds one expert per task, so model size still grows with the tasks. The scalability discussion only shows per-expert parameters, not long-term scaling.
- The gains are mostly incremental, while the method adds substantial architectural and inference complexity. It is unclear whether this tradeoff is preferable to simpler analytic alternatives that also avoid forgetting.

---

> ### Author Rebuttal · Authors · 2026-03-30
>
> We thank the reviewer for recognizing our design choices, evaluation breadth, and improvements over baselines.
>
> ---
>
> **Q1)** Long-term Scalability Analysis
>
> Our method allocates task-specific capacity progressively via lightweight experts (~1.9M params/task), avoiding the large upfront cost of distillation-based methods (e.g., ZSCL: 149.6M from task 1) and growing significantly slower than MoE-Adapters:
>
> | Method | 1 Task | 25 Tasks | 50 Tasks |
> |:---:|:---:|:---:|:---:|
> | ZSCL | 149.6M / 4.20 s/it | 149.6M / 4.20 s/it | 149.6M / 4.20 s/it |
> | MoE-Adapters | 59.8M / 1.58 s/it | 268.6M / 2.59 s/it | 486.1M / 3.77 s/it |
> | **Ours** | **1.9M / 1.31 s/it** | **47.5M / 2.46 s/it** | **95M / 3.94 s/it** |
>
> Key observations:
>
> - **Parameters:** Our method reaches comparable scale to ZSCL only after ~80 tasks; MoE-Adapters already exceeds it at 25 tasks.
> - **Runtime:** At 1–25 tasks, we are substantially faster than ZSCL; at 50 tasks, runtime remains comparable to MoE-Adapters while avoiding their multi-stage inference (DDAS → router → top-K experts).
> - **No auxiliary overhead:** Unlike MoE-Adapters, we require no memory buffers, routers, or reference datasets — growth is restricted to lightweight adapters only.
> - **Sub-task granularity:** Subpopulation experiments (e.g., flamingo within bird) show fine-grained refinements can be handled without proportional expert growth.
>
> Growth is linear but lightweight. In the practical CIL regime (5–50 tasks), our method remains efficient while consistently outperforming both fixed and expanding baselines.
>
> ---
>
> **Q2)** Tradeoff versus RAIL
>
> RAIL is already included as a baseline in the paper, where our method consistently outperforms it (**+6.97% average, +4.36% to +9.70% across 8 benchmarks**).
>
> **Table: Performance Summary**
>
> | Method | C100(10) | C100(20) | TIN(5) | TIN(10) | TIN(20) | E13(5) | E30(4) | X-T | Row Avg |
> |---|---|---|---|---|---|---|---|---|---|
> | RAIL | 76.09 | 76.01 | 71.97 | 72.09 | 72.07 | 81.06 | 78.63 | 81.8 | 76.22 |
> | Ours | 80.6 | 79.32 | 78.95 | 78.31 | 77.42 | 87.08 | 84.88 | 85.63 | 81.52 |
> | Δ% | +5.93 | +4.36 | +9.70 | +8.63 | +7.42 | +7.43 | +7.95 | +4.68 | +6.97 |
>
> **Table: Tradeoff versus RAIL**
> | Aspect | RAIL | Ours |
> |---|---|---|
> | Avg. Performance | 76.22 | **81.52** (+6.97%) |
> | Model Size (TIN(5)) | 169.6M | **161M** |
> | Runtime (50 tasks) | **1.19 s/it** | 3.94 s/it |
> | OOD / X-TAIL | Frozen zero-shot | **Task-specific adaptation** |
>
> **Our model is smaller than RAIL (α ∈ ℝ^{N×c}).** Runtime is the one genuine cost.
>
> **RAIL's simplicity is its ceiling, not just a tradeoff.** It relies on frozen features, a structural limitation no hyperparameter tuning can fix. In X-TAIL and distribution-shift regimes, it falls back to frozen CLIP zero-shot. Our experts adapt representations directly. The gap there is architectural, not tunable.
>
> **Each added component closes a specific failure mode:** confidence scaling handles task coordination (Theorem 3.2); Mahalanobis selection replaces a router without buffers or replay. Neither is arbitrary overhead. (Figs 6-8, Tabs 6-12)
>
> **In CIL literature, +4–10% gains across eight heterogeneous benchmarks are not considered incremental.** The gains are largest precisely where RAIL structurally fails. No replay, no buffers, smaller model, stronger accuracy — the runtime cost is the only concession, and it is justified.
>
> ---
>
> **Q3)** Sensitivity of Confidence + Mahalanobis Combination
>
> Both components play complementary roles: **confidence scaling (τ)** enhances global task coordination during training; **Mahalanobis distance (ζ)** refines expert selection at inference.
>
> Results remain stable across all tested values (Last Acc / Avg Acc):
>
> | | Value | Avg Acc | Last Acc |
> |---|---|---|---|
> | τ | (1.0, 1.0) | 85.57 | 77.88 |
> | τ | **(1.0, 0.8) ✓** | **87.28** | **80.60** |
> | τ | (1.3, 0.7) | 84.37 | 76.27 |
> | ζ | 1.0 | 85.92 | 78.55 |
> | ζ | **0.01 ✓** | **87.28** | **80.60** |
> | ζ | 0.001 | 85.69 | 77.72 |
>
> Table suggest there is no cliff-edge behavior. Gains reflect principled design, not brittle tuning.
>
> ---
>
> ### Summary:
>
> We thank the reviewer for highlighting scalability and complexity concerns. We clarify that linear expert growth is a deliberate tradeoff, enabled by lightweight adapters, resulting in significantly lower growth than proprio MOE approaches. Our additional analysis also shows comparable runtime to the MoE baseline while avoiding routers, replay, and reference datasets. Compared to analytic methods like RAIL, we target a different regime—robustness under distribution shift and task ambiguity—where we consistently achieve stronger performance (e.g., ~+7% average gains). Sensitivity experiments further show that the method is stable across hyperparameters and does not rely on precise tuning. Overall, we clarify that the identified limitation reflects an intentional design tradeoff that enables memory-free, robust continual learning.

---

> > ### Author Rebuttal · Reviewer_fZ6u · 2026-04-04
> >
> > Thank you for the detailed rebuttal. Your response resolves my main concerns. The added long-horizon scaling analysis makes the tradeoff much clearer.

---

> > > ### Author Response · Authors · 2026-04-04
> > >
> > > Thank you for the thoughtful review and for taking the time to consider our rebuttal. We appreciate your feedback and are glad that the additional analysis helped clarify the scalability tradeoff. Thank you again for your support.

---

### Decision · Program_Chairs · 2026-04-30

**Decision:**

Accept (regular)

**Comment:**

This paper receives (444) with three weak-accept reviews. Reviewers raised concerns about long-term scalability, incremental gains, cost analysis, and presentation clarity, which were addressed in the rebuttal. A separate AC-authors clarification on the experiment numbers was resolved. The AC recommends acceptance and encourages the authors to carefully incorporate the rebuttal content into the camera-ready.